# CUT OUT THE ANNOTATOR, KEEP THE CUTOUT: BETTER SEGMENTATION WITH WEAK SUPERVISION

**Sarah M. Hooper**[1],[*] **Michael Wornow**[2], **Ying Hang Seah**[2],
**Peter Kellman**[3], **Hui Xue**[3], **Frederic Sala**[2], **Curtis Langlotz**[4], **Christopher Ré**[2]
[1]Department of Electrical Engineering, Stanford University, Stanford CA, USA
[2]Department of Computer Science, Stanford University, Stanford CA, USA
[3]National Heart, Lung, and Blood Institute, National Institutes of Health, Bethesda MD, USA
[4]Department of Radiology, Stanford University, Stanford CA, USA

## ABSTRACT

Constructing large, labeled training datasets for segmentation models is an expensive and labor-intensive process. This is a common challenge in machine learning, addressed by methods that require few or no labeled data points such as few-shot learning (FSL) and weakly-supervised learning (WS). Such techniques, however, have limitations when applied to image segmentation—FSL methods often produce noisy results and are strongly dependent on which few datapoints are labeled, while WS models struggle to fully exploit rich image information. We propose a framework that fuses FSL and WS for segmentation tasks, enabling users to train high-performing segmentation networks with very few hand-labeled training points. We use FSL models as weak sources in a WS framework, requiring a very small set of reference labeled images, and introduce a new WS model that focuses on key areas—areas with contention among noisy labels—of the image to fuse these weak sources. Empirically, we evaluate our proposed approach over seven well-motivated segmentation tasks. We show that our methods can achieve within 1.4 Dice points compared to fully supervised networks while only requiring five hand-labeled training points. Compared to existing FSL methods, our approach improves performance by a mean 3.6 Dice points over the next-best method.

## 1 INTRODUCTION

Automated image segmentation has seen rapid improvements with recent developments in deep learning (Li et al., 2018; Chen et al., 2017; Milletari et al., 2016). Convolutional neural networks (CNNs) achieve high segmentation performance—but can require large, labeled training datasets. Acquiring training labels is laborious and slow, particularly for medical images where expert segmentation is often required in 3- or 4-dimensions. While large datasets and pre-trained networks exist for natural images, medical image segmentation is a more targeted task, typically requiring new training sets for every imaging modality, scanner type, anatomical structure, and patient population. Such difficulties abound, but the significant impact of improved medical image segmentation motivates tackling these challenges (Hesamian et al., 2019).

Many few-shot learning (FSL) approaches have been proposed to mitigate these difficulties by training networks using only a few labeled examples. For example, data augmentation can reduce the needed amount of labeled data by introducing additional variation into small, labeled training sets through operations such as affine transforms or learned transformations (e.g. deformations learned by GANs) (Zhao et al., 2019; Eaton-Rosen et al., 2018; Çiçek et al., 2016). Many semi-supervised approaches aim to learn a useful representation from unlabeled data then fine-tune on a few manually-annotated images (Chen et al., 2020; Bai et al., 2019; Chen et al., 2019; Chaitanya et al., 2020a). Finally, other approaches aim to transfer knowledge learned from segmenting one class in order to segment a previously unseen class (Shaban et al., 2017; Rakelly et al., 2018; Roy et al., 2020; Ouyang et al., 2020). However, these FSL approaches have limitations. For example, effective data augmentation often requires model- and task-specific transformations that are difficult

---

[*]For correspondence, please contact `smhooper@stanford.edu`

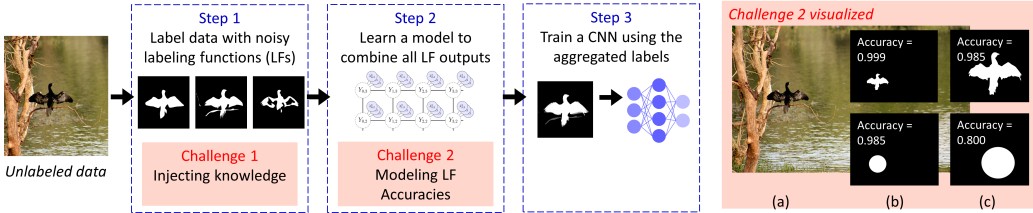

Figure 1: **Left**: Proposed pipeline (blue) and challenges (red). Multiple LFs noisily label data; a label model (LM) then aggregates the LF labels into weak labels that can be used to train a CNN. Two key challenges: how to build LFs that inject knowledge and how to correctly model LF accuracies. **Right**: Second challenge—issues using global accuracy as a metric in the LM: the accuracies of segmentation masks that are clearly correct (b, top) are similar to those that are clearly incorrect (b, bottom). Additionally, segmentation accuracies can be arbitrarily changed by cropping (c).

to generalize to all tasks, and transferring knowledge from one segmentation target to another relies on the segmentations of other classes being present in training datasets. More generally, bringing all of the known domain expertise or structural information to bear is prohibitive.

Weak supervision (WS) is a different approach to training without manual labels that takes advantage of multiple weak labeling sources (Mintz et al., 2009; Craven & Kumlien, 1999; Takamatsu et al., 2012; Gupta & Manning, 2014; Ratner et al., 2019). In many WS workflows, labeling functions (LFs) are crafted to encode the decision-making processes of experts into small functions that programatically label training data. LFs can drastically reduce labeling time, but the resulting labels are often noisy and conflicting. The driving idea behind WS is that the accuracies of and correlations between LFs can be learned without access to ground truth labels, then used to aggregate the LF outputs. This WS workflow enables rapid construction of training sets without manual annotation.

Due to the widespread success of weak supervision methods (Dunnmon et al., 2020; Sheng et al., 2020; Bach et al., 2019), weak supervision is an appealing approach for generating labeled segmentation training sets without relying on hand-labeled data. However, existing WS frameworks do not readily extend to the segmentation setting for the following two reasons:

- **Injecting knowledge.** It is unclear how users can construct multiple segmentation LFs that inject knowledge for a segmentation task at hand—and are suitable for any image.
- **Modeling LF accuracies.** We need to fuse the LFs, but the probabilistic graphical models (PGMs) of existing WS frameworks rely on estimating the accuracies for each labeling function, which is a problematic metric for segmentation: it is not sensitive to small differences at segmentation borders nor robust to arbitrary image crops (Figure 1, right).

We propose a best-of-both-worlds approach that fuses FSL and WS for segmentation tasks. Specifically, we use FSL models as LFs, obviating the need to craft expertise-injecting LFs by hand. To resolve the difficulty in tracking accuracies per LF, which are too coarse of a metric for segmentation, we proposing a new WS PGM. Our PGM introduces conditional terms that "focus" on areas of contention, where LFs disagree, more finely tracking the LF performance in such areas.

We validate our approach on seven medical image segmentation tasks. We show that our method can achieve within 1.4 Dice points of a fully supervised network using only five labeled images. We compare our proposed method to five other few-shot segmentation methods, and find that our method exceeds the next-best by a mean 3.6 Dice points. Finally, we analyze how different FSL methods scale with additional unlabeled training data and compare our PGM to existing WS models.

There are two primary contributions in this work. First, we present a weak supervision pipeline tailored to the segmentation use case which utilizes a small number of labeled images and many unlabeled images to train a high-performing segmentation network. Second, we develop a novel PGM to aggregate noisy segmentation masks and show that the parameters of this model can be learned without access to ground truth labels. We show that our new approach to few-shot image segmentation outperforms many competitive baselines.

## 2   RELATED WORK

**Data augmentation**    Data augmentation has been shown to enable training on few labeled examples, particularly in medical imaging. These augmentations range from simple operations such as affine transforms, elastic transforms, and contrast jitter (Çiçek et al., 2016; Cireşan et al., 2011) to more complex learned operations, such as the use of GANs to learn intensity and spatial deformation transforms to generate synthetic medical images (Chaitanya et al., 2020b).

**Transferring knowledge**    A popular FSL approach enables prediction on previously unseen classes by leveraging representations learned from prior experience (Snell et al., 2017; Sung et al., 2018). Applications to segmentation include Rakelly et al. (2018); Shaban et al. (2017); Dong & Xing (2018). For example, PANet is a proposed approach for few-shot semantic segmentation that utilizes representations for each semantic class from the support set, then labels query images by propagating labels from the support set in embedding space (Wang et al., 2019). Few-shot segmentation was extended specifically to the medical imaging case by Roy et al. (2020), where "squeeze and excite" blocks are used to learn class representations without pretrained networks. However, these approaches rely on the availability of labels of other classes during training.

**Self- and semi-supervised learning**    Self- and semi-supervised methods have been proposed for image segmentation without relying on the availability of other classes during training (Chen et al., 2020; Bai et al., 2019; Chen et al., 2019). Instead, semi-supervised segmentation methods often pretrain on large unlabeled datasets to learn useful representations without any annotations. For example, contrastive learning has been proposed to pretrain CNNs, which are then fine-tuned using a small number of manually segmented images (Chaitanya et al., 2020a). Additionally, superpixel segmentations have been proposed to self-supervise networks, which can be used to segment test images without fine tuning (Ouyang et al., 2020).

In the Appendix, we also discuss conditional random fields and using coarse labels (e.g., bounding boxes, scribbles, and image-level labels) to train segmentation models.

## 3   SEGMENTATION WITH FSL AND WS FUSION

We first give background on WS and state the problem setting. Then, we follow with details of the primary challenges and solutions in each stage of our method. Specifically, we discuss LFs and a new PGM that fuses FSL with WS. We end with a summary of our proposed workflow.

### 3.1   WEAK SUPERVISION BACKGROUND

We build our approach off of principles from WS. We note that WS is an overloaded term in literature. In this work, we refer to WS as the class of approaches in which multiple noisy labeling sources are aggregated into probabilistic training labels without access to ground truth labels (Ratner et al., 2019).

The goal of WS is to best utilize multiple sources of noisy signal (i.e. LFs) to label training data without relying on manual annotation. Each LF takes as input an unlabeled training point and outputs a vote for the label of that training point. These LFs can replace manual training data annotation, however LF votes can be imperfect and conflict with one another. A latent variable model (referred to as a *label model*) can be used to model these observed LFs and their relationship to the latent ground truth label. In Figure 2a, we show a simple model where five LFs each vote on the label for a single unobserved ground truth label. Importantly, the parameters of this model need to be estimated without access to ground truth annotation. The learned label model can be then used to produce probabilistic training labels to train a downstream network, referred to as an *end model*.

Building off of this past work, our weak supervision pipeline proceeds in three steps: (1) noisy labels are generated by multiple LFs, (2) the noisy labels are aggregated into weak labels by a probabilistic graphical model (PGM), and (3) the weak labels are used to train a segmentation model (Figure 1).

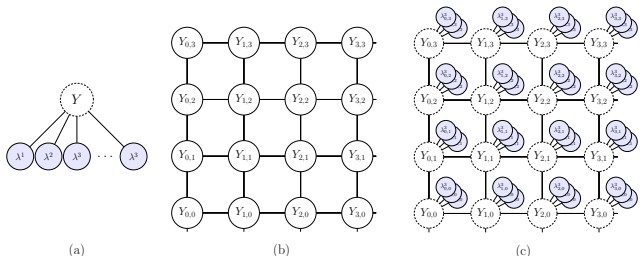

Figure 2: Example latent variable models. (a) A simple PGM with one latent variable $Y$ and several observed labeling functions $\lambda^i$. (b) Conventional segmentation grid. (c) The PGM used in this work, where all nodes in a grid of latent variables are weakly labeled by labeling functions.

### 3.2 PROBLEM SETTING

We are given a training set of $n$ images, $\mathbf{X} = [X_1, X_2, ..., X_n]$, where $X_i \in \mathbb{R}^{I \times J \times K \times T}$ is a tensor of $P$ pixels. $I, J, K,$ and $T$ represent the dimensions of image $X_i$, where $K > 1$ for volumetric images and $T > 1$ for time series images. For each image $X_i$ there is a segmentation mask $Y_i \in \{\mathbf{c}\}^{I \times J \times K \times T}$, where $\{\mathbf{c}\}$ is the set of class labels, yielding the set $\mathbf{Y} = [Y_1, Y_2, ..., Y_n]$ of all *unobserved* ground truth segmentation masks. We have access to $m$ labeling functions that provide noisy segmentation masks $\lambda_i^l \in \{\mathbf{c}\}^{I \times J \times K \times T}$ for each $X_i$, where $l \in \{1, 2, ..., m\}$. From these observed sources, we seek to generate the set of weak labels $\hat{\mathbf{Y}}$ to estimate $\mathbf{Y}$. $\hat{\mathbf{Y}}$ provides labels for a large dataset, and we can use these to train any end model (e.g., a segmentation CNN).

We tackle the binary segmentation case, $\{\mathbf{c}\} = \{-1, 1\}$; simple multiclass extensions are possible.

**Model Structure**   The (unobserved) ground truth mask $Y$ and the (observed) sources $\lambda$ are jointly modeled by an undirected graphical model (Koller & Friedman, 2009; Wainwright & Jordan, 2008) with graph $G = (V, E)$. Since $Y$ is unobserved, this is a latent-variable model. We use conventional structures for each component and combine them in a natural way:

- In a standard weak supervision approach, the $m$ weak sources are connected to a ground-truth label $Y$ (Fig. 2a),
- The segmentation mask $Y$ is a multi-dimensional grid over the pixels (Fig. 2b),
- Our full model has one copy of the WS graph for each pixel in the grid (Fig. 2c).

Our goal is to construct effective LFs and to complete the model specification above by defining the potentials associated with the edges in $E$ and the corresponding parameters.

### 3.3 INJECTING KNOWLEDGE WITH FSL-BASED LABELING FUNCTIONS

In Step 1 of our workflow (Figure 1), we use LFs to generate multiple noisy segmentation masks for each unlabeled image. These LFs transfer the decision-making processes about labeling data into functions that can be run programmatically over unlabeled training data.

**Challenge**   LF development typically involves writing short programs expressing heuristics (Ratner et al., 2018) or external knowledge base lookups (Mintz et al., 2009; Takamatsu et al., 2012). However, even with computer vision expertise, it is difficult for a new user to write segmentation heuristics as LFs for each new task at hand, as is required of users from existing WS tools.

**Solution**   We propose FSL models as a unified way to inject labeling knowledge into LFs. In this work, we use one-shot learning: we train a CNN from a single labeled data point for each LF. However, we note that any few-shot model could be used as a labeling function. Fusing FSL and WS allows us to build upon the strengths of each:

- From WS, we build from the idea that we can track the noise in multiple labeling sources to aggregate labels. We improve upon existing WS tools by showing that few-shot networks—

which obviate the need for users to write their own LFs, as is required by existing WS frameworks—are flexible and strong LFs in a WS segmentation pipeline.

- From FSL, we take ideas for how to train networks with very few labeled images. We improve upon these methods by tracking the noise in each FSL model with WS.

Our FSL-based LFs depart from the domain-expert LF development approach, which is often repeated for each task. Instead, the same process is followed for all tasks explored here: five images are annotated by hand, each of which are used to train a one-shot model. The one-shot models function as LFs that weakly label the rest of the training set. In Section A.2 of the Appendix, we discuss other segmentation LFs we considered.

### 3.4 Modeling accuracies with contention-based probabilistic model

In Step 2 of our workflow (Figure 1), we aggregate the noisy segmentation masks produced by the LFs into probabilistic training labels.

**Challenge**   To aggregate the outputs of the LFs, we need to define the graph structure $G$ (given in Figure 2c) and the potentials/parameters associated with the edges in $G$. However, it is not obvious how to do so. On one extreme, separate parameters for each edge in the model in Figure 2c easily results in millions of parameters to be learned. At the other extreme, treating each pixel as independent of all other pixels (as in separate copies of Figure 2a) results in the problems summarized in Figure 1 (right): the parameters tend to be equivalent, since all LF accuracies are high and similar, and can be changed by arbitrary crops.

**Solution**   Our approach hits a sweet spot. It exploits a notion of *LF contention*, trading off between the naive model and the highly-complex model that has no parameter-tying.

*Model Potentials, Parameters, and Contention*

The latent variable model in Figure 2a expresses the relationship between the sources and unobserved ground truth label. The potentials are typically those from a binary Ising model (Bach et al., 2017; Ratner et al., 2019; Varma et al., 2019; Fu et al., 2020). The density function is defined as,

$$p(Y, \lambda_1, ..., \lambda_m) = \frac{1}{Z} \exp\left(\theta_Y Y + \sum_i \theta_i \lambda_i Y\right) \tag{1}$$

where $\theta_Y \in \mathbb{R}$ and $\theta_i \in \mathbb{R}$. Intuitively, the $\theta_i$ parameters control the accuracies of the sources. If $\theta_i = 0$, then $\lambda_i$ is independent of $Y$—and thus can only output a random guess of $Y$. As $\theta_i$ increases, $\lambda_i$ and $Y$ vote together more often, and so $\lambda_i$ is more accurate. For this reason, it is crucial to estimate the parameters of this model; naively combining the noisy labels will over-weight the inaccurate sources.

In our segmentation structure (Fig. 2c), there are many copies of this model. Using unique parameters for each copy yields an intractably complex model. Tying parameters naively (i.e., each LF has the same accuracy across all pixels) is too coarse—even a poor LF will typically be very accurate across the entire image. To tackle this, we introduce a notion of LF contention into our density function. Specifically, we use indicator variables that depend on LFs agreeing or disagreeing on a given pixel. Intuitively, this will allow the model to learn a certain set of parameters when the LFs are all agreeing (e.g. image background) vs. when the LFs have some disagreement (e.g. segmentation border). By tying parameters based on LF agreements, this distribution allots additional model parameters to areas of contention without introducing too many free parameters to model the grid-structured PGM. Moreover, this approach enables differential model parameters to be learned even when global LF accuracies are highly similar, and is robust to arbitrary image crops.

We refer to a set of LF agreement conditions over some LF subset $D$ as $C_D$. $C_D$ exhaustively enumerates each possible combination of LF agreements and disagreements, $C_i$. For example, the $C_D$ for $D = \{\lambda_1, \lambda_2, \lambda_3\}$ is given by

$$C_D = \{C_1, C_2, C_3, C_4\}$$

$$C_1 = [\lambda_1 \neq \lambda_2, \lambda_1 \neq \lambda_3, \lambda_2 = \lambda_3], C_2 = [\lambda_1 \neq \lambda_2, \lambda_1 = \lambda_3, \lambda_2 \neq \lambda_3]$$

$$C_3 = [\lambda_1 = \lambda_2, \lambda_1 \neq \lambda_3, \lambda_2 \neq \lambda_3], C_4 = [\lambda_1 = \lambda_2, \lambda_1 = \lambda_3, \lambda_2 = \lambda_3]$$

A technical detail we exploit for learning: we always leave two LFs out of the contention set $D$ to ensure sufficient conditional independence for parameter recovery, as we describe next. The set of these left-out LFs is $I$; if there are $m = 5$ LFs, then $I = \{\lambda_4, \lambda_5\}$.

Formally, we define the density of the contention-based model as

$$p(Y_i, \lambda_i^1, ..., \lambda_i^m) =$$

$$\frac{1}{Z} \exp \left( \underbrace{\sum_{d=1}^{|C|} \mathbb{1}\{C_d\}}_{\text{Contention term}} \Big( \underbrace{\sum_{p=1}^{P} \big( \theta_{Y_d} Y_i^p + \sum_{k \in D \cup I} \theta_{\lambda_d^k} \lambda_i^{p,k} Y_i^p \big)}_{\text{Weak supervision}} + \underbrace{\sum_{(p,q) \in E} \theta_{Y Y_d} Y_i^p Y_i^q}_{\text{Segmentation}} \Big) \right) \quad (2)$$

where each canonical parameter $\theta_i \in \mathbb{R}$. In practice we only sum over a single element of $D$, which is an equivalent but minimal distribution. enabling backwards mapping as described below.

*Learning & Inference*

Given observed values of LFs over a set of images, we wish to learn the parameters $\theta$ from (2). Critically, this must be done without relying on ground truth segmentation masks—so that only the $\lambda$ variables, and not $Y$, are observed.

We use an extension of Fu et al. (2020)'s *method-of-moments*-based approach. For the model (1), $\mathbb{E}[\lambda_i \lambda_j] = \mathbb{E}[\lambda_i Y] \mathbb{E}[\lambda_j Y]$. The left-hand side can be estimated (as we observe the LFs); a system with three such equations, for the pairs $(i, j), (i, k), (j, k)$, determines $\mathbb{E}[\lambda_i Y]$ up to sign. These terms are the *mean* parameters of the model, and under mild technical conditions, they determine the canonical parameters $\theta$ (Wainwright & Jordan, 2008). Our extension incorporates the contention term into the decomposition. The full details are in the Appendix A.3.

After solving for the label model parameters, our goal is to compute the marginal probability for each pixel $Y_i^p$ given the observed values of the labeling functions for all pixels in our graph. Various methods have been proposed for performing inference over a large and cyclical graphical model. Here, we approximate the exact conditional using a common graph-cut algorithm. We learn the parameters and compute the unary potentials at each pixel, treating each pixel as independent at this stage. The max-flow/min-cut algorithm proposed by Boykov & Kolmogorov (2004) is used to efficiently compute the final segmentation mask considering pairwise interaction between all neighboring pixels. However, we note that many approximate inference algorithms could be used as well, including approximate belief propagation algorithms that would return probabilistic labels.

### 3.5 SUMMARY OF WORKFLOW

Taking the solutions developed in Sections 3.3 and 3.4, we put the pieces together and describe the full workflow of our proposed approach (Figure 3).

- Step 1. Each one of the five manually annotated images is used to train a one-shot CNN. With five manually annotated images, there are five one-shot CNNs. Each of these one-shot CNNs is a labeling function which labels all unlabeled images in the training set.
- Step 2. A PGM is used to aggregate the five noisy segmentation masks for each image in the unlabeled training set. The parameters of this PGM are learned for each dataset without using ground truth segmentation masks.
- Step 3. A CNN is trained using the estimated segmentation masks output from the PGM, then fine-tuned with the few hand-labeled images.

## 4 EXPERIMENTS

We describe our experimental setup in Section 4.1. The primary goal of our evaluation is to show that our proposed weak supervision method can achieve high performance across a range of medical image segmentation tasks (Section 4.2). Secondly, we aim to more deeply understand the benefits of the proposed method compared to other methods (Section 4.3).

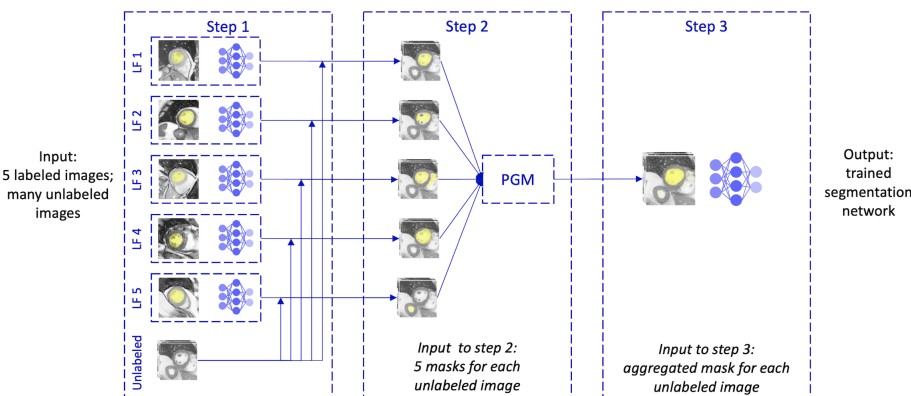

Figure 3: Details of proposed workflow. In Step 1, each manually-annotated image is used to train a one-shot model, which in turn produces a noisy label for all unlabeled training images. In Step 2, the outputs from the FSL LFs are aggregated via a PGM, the parameters of which are learned without access to any ground truth labels . In Step 3, the PGM outputs are used to train an end model.

## 4.1 EXPERIMENTAL SETUP

**Datasets**  We select public segmentation datasets to evaluate our proposed method.  We chose datasets to include varied imaging modalities, body regions, and structures.

- **ACDC.** 100 volumetric short-axis cardiac magnetic resonance (CMR) cine images available for download, totaling 951 2D axial slices. The right ventricle, the left ventricle endocardium, and the myocardium segmentations were provided (Bernard et al., 2018).
- **Abdominal CT.** 43 contrast-enhanced abdominal computed tomography (ACT) scans, totaling 10,235 2D axial slices. We use the segmentations provided for the spleen, esophagus, and liver (Gibson et al., 2018; Roth et al., 2016; 2015; Clark et al., 2013).
- **BRATS.** 210 head magnetic resonance (HMR) scans, totaling 31,775 axial slices. We use the flair HMR sequence and segment any region positive for tumor (Menze et al., 2014; Bakas et al., 2017; 2018).

**Baselines**  We compare our proposed approach to five other few-shot segmentation methods. We select the baselines that have similar inputs (a few labeled images and many unlabeled images) and outputs (a trained segmentation neural network) to our proposed approach, but use a distinct method, and have published implementations by the authors.  In each case, we use the same unlabeled and five labeled images we use in our proposed approach to ensure the training datasets are comparable.

- **Data augmentation.**  We compare against a data augmentation method that trains generative adversarial networks (GANs) using unlabeled images, then transforms the labeled images using the trained GANs to generate a larger labeled training set ("GAN DA") (Chaitanya et al., 2020b).
- **Knowledge transfer.**  We compare against PANet (Wang et al., 2019), a FSL approach to semantic segmentation developed for natural images.  We also compare against Squeeze and Excite ("S&E"), a few-shot approach developed for medical images (Roy et al., 2020).
- **Semi- and self-supervised learning (SSL).** We compare against a semi-supervised method that pretrains using a contrastive loss over many unlabeled images, then fine-tunes on a few manually-annotated images ("CL") (Chaitanya et al., 2020a).  We also compare against SSL-ALPNet, which utilizes a self-supervised superpixel segmentation task then uses the learned representations to segment new classes without fine-tuning (Ouyang et al., 2020).
- **Traditional supervision.**  We train a model using only the five labeled reference images as the lower bound of expected performance ("FS-5").  We also train a fully supervised network using all ground truth segmentations of all training data as the upper bound of expected performance ("FS-all").

Table 1: Performance of proposed method and baselines.

| Method | Cardiac MR | | | | Ab. CT | | BRATS |
| | Endo. | Myo. | RV | Liver | Spleen | Esophagus | Tumor |
| --- | --- | --- | --- | --- | --- | --- | --- |
| FS-5 | 78.8 (1.2) | 55.5 (2.4) | 52.3 (8.7) | 92.0 (0.7) | 92.7 (0.3) | 48.2 (8.3) | 77.1 (0.5) |
| GAN DA | 90.8 (2.7) | 76.3 (0.4) | 76.8 (0.4) | 86.9 (1.9) | 84.0 (5.1) | 54.9 (0.8) | 73.9 (4.1) |
| PANet | 62.0 (3.3) | 33.2 (8.2) | 42.3 (1.2) | 58.0 (1.7) | 23.0 (1.0) | 25.7 (17.3) | N/A |
| S&E | 46.9 (3.3) | 21.1 (7.9) | 25.2 (2.3) | 24.0 (1.7) | 06.7 (3.2) | 00.8 (0.6) | N/A |
| SSL-ALPNet | 83.0 (1.4) | 38.6 (1.3) | 70.7 (1.7) | 77.4 (0.3) | 41.4 (0.7) | 36.6 (4.5) | 13.3 (1.4) |
| CL | 90.5 (1.0) | 77.2 (1.1) | 81.1 (2.0) | 91.8 (0.3) | 87.8 (2.8) | 61.8 (3.1) | **78.4 (0.8)** |
| Proposed | **94.7 (0.5)** | **81.0 (1.2)** | **87.2 (3.2)** | **93.2 (0.5)** | **93.1 (0.4)** | **69.2 (1.7)** | 77.4 (0.7) |
| FS-all | 96.1 (0.3) | 87.0 (0.5) | 93.6 (0.1) | 96.1 (0.2) | 94.5 (0.3) | 75.6 (2.1) | 83.5 (0.5) |

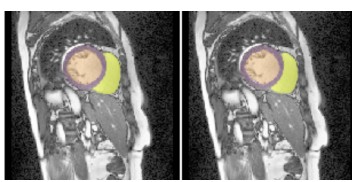 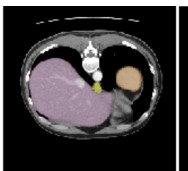 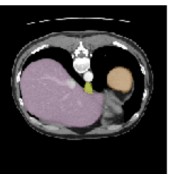

Figure 4: Example results. The left of each pair is the ground truth, the right of each pair is produced by the proposed method. Left: orange is endocardium, pink is myocardium, yellow is right ventricle. Right: orange is spleen, pink is liver, yellow is esophagus.

**Training and Evaluation** Each dataset is split into 60% training, 20% validation, and 20% testing. Image preprocessing is described in the Appendix A.4. For the LFs, FS-5, FS-all, and our proposed approach, we train a 2D U-Net (Ronneberger et al., 2015) to minimize the binary cross entropy loss using the Adam optimizer with learning rate set to 1e-3. Batch size was set to 16. Random affine transforms, color jitter, and elastic transforms were used during training for all models except FS-5. We note that the performance of FS-all will be lower than that reported in public leaderboards due to our simple end model. Because we aim to show how well our proposed method utilizes unlabeled data and a small set of labeled data, we keep the end model simple to be comparable to baselines. More complex networks could easily be substituted into Step 3.

The Dice coefficient, $Dice(Y_i, Y_j) = \frac{2|Y_i \cap Y_j|}{|Y_i| + |Y_j|}$, is used to evaluate networks. Each network was trained with three different random seeds. The mean (standard deviation) Dice scores are reported.

## 4.2 PERFORMANCE OF PROPOSED METHOD

### 4.2.1 COMPARISON TO BASELINES

The Dice scores of our proposed approach and baselines are shown for the seven segmentation tasks in Table 1; visualizations of the segmentations are in Figure 4 and in the Appendix Figure A.1. Despite using only five labeled images, we obtain near fully supervised performance on the endocardium, liver, and spleen, achieving on average within 1.9 points of the fully supervised network. On the more difficult segmentation tasks, we still see boost compared to existing approaches, though there is larger gap between FS and WS performance. This is to be expected, since the segmentation of smaller structures is more difficult. Still, we see strong agreement in the shape and location of the WS segmentations compared to the ground truth segmentations.

### 4.2.2 INFLUENCE OF SELECTED LABELED IMAGES

We evaluate how our results change depending on the five manually-annotated images. The original set of manually-annotated images were selected randomly. We select three additional sets of reference images and re-run our proposed method; all sets are mutually exclusive. The results from each new set (run with the same random seed) are shown in Table 2. In general, we see good agreement across different randomly-selected labeled images. For all datasets except the right ventricle and the esophagus, we see standard deviations of less than 1.5 Dice points. Selecting images to label from an unlabeled dataset to improve end model results would be an interesting direction for future work.

Table 2: Performance of proposed method with different sets of labeled images.

| Labeled image set | Cardiac MR | | | | Ab. CT | | BRATS |
|---|---|---|---|---|---|---|---|
| | Endo. | Myo. | RV | Liver | Spleen | Esophagus | Tumor |
| Original set | 94.9 | 79.7 | 83.9 | 93.7 | 92.9 | 70.4 | 77.4 |
| Set 1 | 93.4 | 81.4 | 89.9 | 95.6 | 93.1 | 73.2 | 75.4 |
| Set 2 | 94.9 | 81.8 | 89.5 | 94.5 | 91.1 | 51.6 | 76.5 |
| Set 3 | 95.1 | 82.1 | 91.7 | 93.8 | 91.7 | 66.1 | 75.3 |

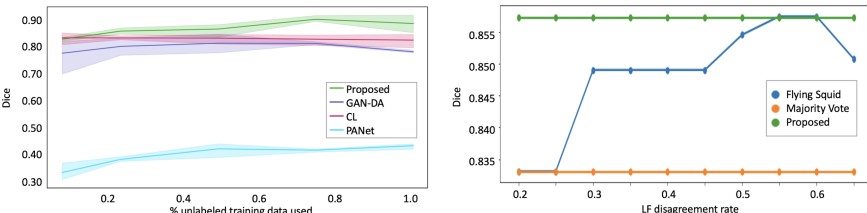

Figure 5: Comparisons for different amounts of unlabeled data (left) and image crops (right).

### 4.3 TRADEOFFS WITH OTHER APPROACHES

**Performance scaling with amount of unlabeled data**  Since unlabeled data is much easier to collect than manually-annotated data, FSL approaches that are able to effectively utilize unlabeled data may have an advantage when deployed to real-world applications. In Figure 5 (left), we show the performance of our proposed approach, the highest performing knowledge transfer approach (PANet), the data augmentation network (GAN-DA), and the highest performing self- or semi-supervised network (CL) as more unlabeled data is provided during training for the CMR right ventricle task. We observe that the performance separation between WS and other approaches is larger with more unlabeled data. In the low data regime, WS, CL, and GAN-DA perform similarly.

**Label model performance**  To show the advantages of the proposed segmentation LM versus existing WS models, we conduct an experiment with synthetic datasets. We generate a dataset according to Equation 2, and pad this dataset with negatively labeled examples. This process mimics cropping an image around the ROI. We show the performance of our proposed LM, majority vote (MV), and a leading existing WS framework (Flying Squid, Fu et al. (2020)) in Figure 5 (right). We see our proposed model is unaffected by background cropping (or equivalently, LF disagreement rate) while the existing WS tool has varying performance as a function of arbitrary crops. We report the Dice score of MV, the proposed LM, and Flying Squid in the Appendix A.4. Finally, in Appendix A.4, we also include an experiment on the generalization performance of the FSL models.

## 5 CONCLUSION

Automated medical image segmentation is a difficult problem. There are numerous imaging modalities, views, and structures to segment and each application demands high performance, leading to extensive labeling costs from experts. We present a general approach to training segmentation networks with limited hand labels, fusing FSL and WS to build upon the benefits of each. This approach is flexible, capable of utilizing various forms of noisy signal to weakly label training data, which can in turn be used to train any downstream model. Our method is compatible with future improvements in segmentation methods: new few-shot approaches can be used as LFs in Step 1, and more complex segmentation networks or loss functions can be used in Step 3. For example, unsupervised pre-training could be used to initialize the one-shot networks used as LFs, such as the methods proposed in Chaitanya et al. (2020a). In fact, any of the baselines we compare against in this work could be used as LFs. Finally, we have shown that the proposed approach to training segmentation networks with few labels outperforms many competitive baselines on a range of different imaging modalities and anatomical structures.

ACKNOWLEDGMENTS

Sarah Hooper is supported by the Fannie and John Hertz Foundation, the National Science Foundation Graduate Research Fellowship under Grant No. DGE-1656518, and as a Texas Instruments Fellow under the Stanford Graduate Fellowship in Science and Engineering. We gratefully acknowledge the support of NIH under No. U54EB020405 (Mobilize), NSF under Nos. CCF1763315 (Beyond Sparsity), CCF1563078 (Volume to Velocity), and 1937301 (RTML); ONR under No. N000141712266 (Unifying Weak Supervision); the Moore Foundation, NXP, Xilinx, LETI-CEA, Intel, IBM, Microsoft, NEC, Toshiba, TSMC, ARM, Hitachi, BASF, Accenture, Ericsson, Qualcomm, Analog Devices, the Okawa Foundation, American Family Insurance, Google Cloud, Swiss Re, Total, the HAI-AWS Cloud Credits for Research program, the Stanford Data Science Initiative (SDSI), and members of the Stanford DAWN project: Facebook, Google, and VMWare. The Mobilize Center is a Biomedical Technology Resource Center, funded by the NIH National Institute of Biomedical Imaging and Bioengineering through Grant P41EB027060. The U.S. Government is authorized to reproduce and distribute reprints for Governmental purposes notwithstanding any copyright notation thereon. Any opinions, findings, and conclusions or recommendations expressed in this material are those of the authors and do not necessarily reflect the views, policies, or endorsements, either expressed or implied, of NIH, ONR, or the U.S. Government.

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

# A  APPENDIX

## A.1  EXTENDED RELATED WORK

**Coarse labels**  Coarser forms of annotation per image can be used to train segmentation models with less manual labor. For example, bounding boxes, scribbles, and class labels can be used to provide supervision for segmentation models (Sakinis et al., 2019; Xu et al., 2015; Khoreva et al., 2017). This approach is sometimes referred to as "weak supervision" in the segmentation literature. We note that this technique is different than the WS approach presented here, as it still requires each training sample to be annotated and there is not a unified model to aggregate multiple noisy labeling sources. We refer to weak supervision as the process by which multiple noisy segmentation masks are generated and aggregated automatically.

**Conditional random fields**  Traditional approaches to modeling segmentation masks for images use conditional random fields (CRFs). These distributions often utilize unary potentials over each pixel and pairwise potentials between pixels. Many methods were proposed for defining the unary potentials—including engineering features over the color, location, and texture of regions in the image (Shotton et al., 2009)—as well as pairwise interactions between pixels, such as how smooth segmentations were expected to be (Kohli et al., 2009; Krähenbühl & Koltun, 2011). Additional work was devoted to performing inference over the large CRFs, developing new methods for approximate inference via belief propagation and graph cuts (Kohli et al., 2007; Boykov et al., 2001; Boykov & Kolmogorov, 2004).

We note that the graph structure we use (Figure 2c) bears resemblance to those used in past work in CRFs. However, in most traditional CRF frameworks, the parameters of the PGM are learned or tuned from labeled training data. Here, we work with a latent ground truth variable $Y$ and generate probabilistic labels for each pixel by learning how to aggregate LF votes without access to ground truth data.

## A.2  ALTERNATIVE LABELING FUNCTIONS

We experimented with many different types of LFs in the course of this work. We focus on the one-shot models used in this work because they were general (i.e., can be applied to any task without requiring user-modification) and produced strong results. However, for certain tasks, additional LFs may be included in Step 1 to provide additional signal to the PGM. For example,

- **Physical models.** For segmentation problems where structures in all images are arranged similarly, physical models (e.g., atlases) can be registered to the unlabeled images and provide weak labels via label propagation.
- **Pre-trained networks.** If the new segmentation task is similar to commonly-studied tasks (e.g. in natural image segmentation), off-the-shelf networks may be deployed.
- **Image features.** For simpler tasks, latent image features (e.g. color, intensity) can provide weak segmentation signal. These features can also be utilized via traditional CV segmentation approaches such as active contour, random walker, and watershed segementation.
- **Clustering.** Unsupervised segmentation methods such as SLIC, which clusters images in XYZ-color space, can be easily implemented for tasks that have distinct boundaries.

## A.3  LEARNING MODEL PARAMETERS

We rely on a technical fact: our proposed distribution is minimal, so that finding the canonical parameters $\theta$ can be accomplished by finding the mean parameters of the model: $E[\mathbb{1}\{C_d\}Y_i^p]$, $E[\mathbb{1}\{C_d\}Y_i^pY_i^q]$, and $E[\mathbb{1}\{C_d\}\lambda_i^{p,k}Y_i^p]$ and performing the mean-to-canonical parameter mapping. See Wainwright & Jordan (2008) for more details. We take $E[\mathbb{1}\{C_d\}Y_i^p]$ and $E[\mathbb{1}\{C_d\}Y_i^pY_i^q]$ as inputs, representing the conditional class balance of all pixels and the conditional rate of agreement of neighboring pixels, respectively. We note that these values can be estimated from a validation set, tuned, or estimated from the observed values of the labeling functions. The remaining mean parameters to be estimated are then $E[\mathbb{1}\{C_d\}\lambda_i^{p,k}Y_i^p]$. For conciseness, we will refer to $Y_i^p$ as $Y$ and $\lambda_i^{p,k}$ as $\lambda^k$ in the remainder of this section.

The mean parameter $E[\mathbb{1}\{C_d\}\lambda^k Y]$ can be interpreted as the scaled accuracy of labeling function $S^k$ under condition $C_d$, $a_d^k$. That is, rather than being between 0 and 1, it is scaled to be between $-1$ and 1:

$$a_d^k = E[\mathbb{1}\{C_d\}\lambda^k Y] = P((\lambda^k = Y) \cap (C_D)) - P((\lambda^k \neq Y) \cap (C_D)). \tag{3}$$

This mean parameter cannot be directly estimated (i.e., from samples of $\lambda$ and $Y$) as we do not observe the ground truth label $Y$. Instead, building off of the triplet method for parameter recovery presented in Fu et al. (2020), the conditional accuracy of each labeling function can be efficiently computed in closed-form using a system of equations.

We use the following decomposition to factor an observable quantity into the product of two conditional accuracies $a_d^j$ and $a_d^k$. We take $j$ to be in the contention set $C$ and $k$ to be in the non-contention part $I$. The decomposition holds due to the fact that these terms are uncorrelated, which follows from the conditional independence of the underlying $\lambda$ variables.

$$
\begin{aligned}
a_d^j a_d^k &= E[\mathbb{1}\{C_d\}\lambda_i^{p,j} Y_i^p] E[\mathbb{1}\{C_d\}\lambda_i^{p,k} Y_i^p] \\
&= E[\mathbb{1}\{C_d\}\lambda_i^{p,j} Y_i^p \mathbb{1}\{C_d\}\lambda_i^{p,k} Y_i^p] \\
&= E[\mathbb{1}\{C_d\}\lambda_i^{p,j} \lambda_i^{p,k}].
\end{aligned}
$$

Having access to conditionally independent labeling functions is precisely why we selected two LFs and kept them in the set $I$, which does not overlap with the contention set $D$.

With the equation above, we can observe all of the terms such as $\mathbb{1}\{C_d\}\lambda_i^{p,j}\lambda_i^{p,k}$, so that we can empirically estimate the final term. If we form triplets of such equations, for the pairs $(j,k),(j,\ell),(k,\ell)$, where $j \in D$ and $k,\ell \in I$, then we can use the following system of equations to solve for the $a_d^j, a_d^k, a_d^\ell$, up to sign. Specifically, the system is

$$
\begin{aligned}
a_d^j a_d^k &= E[\mathbb{1}\{C_d\}\lambda_i^{p,j}\lambda_i^{p,k}], \\
a_d^j a_d^\ell &= E[\mathbb{1}\{C_d\}\lambda_i^{p,j}\lambda_i^{p,\ell}], \\
a_d^k a_d^\ell &= E[\mathbb{1}\{C_d\}\lambda_i^{p,k}\lambda_i^{p,\ell}].
\end{aligned}
$$

The magnitude of the solution is for $j$

$$|a_d^j| = \sqrt{\frac{E[\mathbb{1}\{C_d\}\lambda_i^{p,j}\lambda_i^{p,k}] \times E[\mathbb{1}\{C_d\}\lambda_i^{p,j}\lambda_i^{p,\ell}]}{E[\mathbb{1}\{C_d\}\lambda_i^{p,k}\lambda_i^{p,\ell}]}}.$$

In practice, we use the empirical versions of each of the expectation terms. In general, as long as the accuracies are better than random, the sign is positive, so that we fully recover $a_d^j$. Even in cases where the signs of some accuracies are negative, it is still possible to recover in a range of conditions, using the technique detailed in Ratner et al. (2019).

This procedure can be done for any of the LFs and choices of the contention. All that is required is to select valid triplets. By design, there is only one possible valid grouping of triplets: each labeling function in $D$ upon which the conditions rely is grouped with the two labeling functions in $I$ upon which no conditions rely. We take the median conditional accuracy estimate for each labeling function in $I$. The division of labeling functions into $I$ and $D$ is chosen to maximize the number of LF disagreements caught by the conditionals in the dependent set.

With this we have recovered all of the mean parameters. To obtain the canonical parameters—the $\theta$'s in (2)—we use the mean-to-canonical mapping, as detailed in Wainwright & Jordan (2008).

## A.4 Additional Experimental Results

**Image preprocessing** The CMR images were corrected with an N4 bias field correction and normalized with histogram equalization. The ACT images were preprocessed with a CT window (window level = 50, window width = 400) and each axial slice was resized to 224x224, then normalized to the range [0,1]. The BRATS dataset was resized such that each axial slice was 224x224, then normalized to the range [0,1].

**Ablations** We explore the effect of each step of our pipeline in Table 1 below. We note that these are the Dice scores reported over the validation sets because the test set is never processed by the LFs.

- **Effect of LF aggregation.** We report the score of each of our one-shot LFs and compare it to the Dice score of the aggregated LFs.

- **Effect of aggregation rule.** We compare different aggregation rules on our datasets: the majority vote of the labeling functions, our proposed label model, and Flying Squid (Fu et al. (2020)). We note that the current implementation of Flying Squid was unable to perform inference over the large number of pixels in the ACT and BRATS datasets.

- **Effect of training end model.** We show the impact of training an end model with the weak labels compared to the weak labels alone.

- **Effect of fine-tuning.** We show the impact of fine-tuning the end model with the small number of hand-labeled examples.

Table 1: Ablations.

| Method | Cardiac MR | | | | Ab. CT | | BRATS |
| | LV Endo | LV Epi | RV | Liver | Spleen | Esophagus | Tumor |
|---|---|---|---|---|---|---|---|
| Labeling functions | | | | | | | |
| LF 1 | 89.0 (0.7) | 75.3 (0.3) | 71.4 (1.7) | 91.8 (0.2) | 86.2 (0.6) | 22.2 (3.0) | 71.7 (1.8) |
| LF 2 | 81.5 (1.5) | 45.3 (3.2) | 55.1 (2.6) | 92.5 (0.1) | 70.6 (3.3) | 31.2 (1.9) | 68.8 (3.1) |
| LF 3 | 84.9 (0.5) | 59.7 (0.8) | 62.9 (1.1) | 92.8 (0.2) | 88.3 (1.2) | 25.3 (3.3) | 73.1 (4.0) |
| LF 4 | 90.5 (0.6) | 74.1 (0.6) | 80.8 (1.8) | 83.7 (0.9) | 72.1 (1.0) | 30.0 (1.2) | 73.8 (2.8) |
| LF 5 | 86.2 (1.6) | 58.8 (0.5) | 61.4 (1.9) | 93.0 (0.6) | 86.7 (0.3) | 46.5 (1.3) | 70.8 (5.0) |
| Aggregation rules | | | | | | | |
| Majority vote | 92.4 (0.4) | 76.8 (0.8) | 79.1 (1.6) | 94.4 (0.3) | 91.2 (0.1) | 42.1 (0.4) | 77.0 (1.0) |
| Flying squid | 92.3 (0.5) | 76.7 (0.6) | 79.0 (2.0) | - | - | - | - |
| Proposed PGM | 92.8 (0.2) | 77.7 (0.4) | 82.3 (1.2) | 94.5 (0.2) | 90.7 (0.5) | 45.1 (2.8) | 77.3 (0.6) |
| End model | | | | | | | |
| End model | 94.0 (0.2) | 79.3 (0.5) | 85.0 (0.9) | 95.0 (0.2) | 94.1 (0.2) | 51.7 (0.0) | 78.0 (0.8) |
| Fine tuning | | | | | | | |
| End model + FT | 94.4 (0.1) | 82.4 (0.3) | 87.5 (1.5) | 95.7 (0.1) | 94.2 (0.3) | 59.5 (0.4) | 77.7 (0.3) |

**Additional qualitative results** In Figure A.1, we show qualitative visualizations of the segmentations described in Section 4.2.1.

**Generalization performance** For methods that use fewer training labels, we hypothesized that the final CNNs may be less robust to natural image variations than fully supervised networks. We assessed the models' abilities to generalize to new test sets for two tasks. While the new datasets are of the same structure, view, and modality as the originals, they represent different patient populations and scanners. As an additional CMR test set, we collected 100 CMR images with endocardium segmentations through collaboration with the NIH. As an additional ACT test set, we use images and liver segmentations from the CHAOS challenge (Kavur et al., 2020a;b; 2019).

Table 2: Generalization performance.

| | Endo. | Liver |
|---|---|---|
| PANet | +3.76% (4.93%) | +6.35% (3.52%) |
| GAN DA | -15.30% (3.01%) | +2.07% (1.32%) |
| CL | -11.22% (3.98%) | -0.69% (0.97%) |
| Proposed | -5.38% (0.87%) | +2.48% (0.66%) |
| FS-all | -3.94% (0.37%) | +0.38% (0.27%) |

We compare the performance of the trained models on these new test sets and report the change in performance compared to the original test set in Table 2 of the Appendix. We compared the best

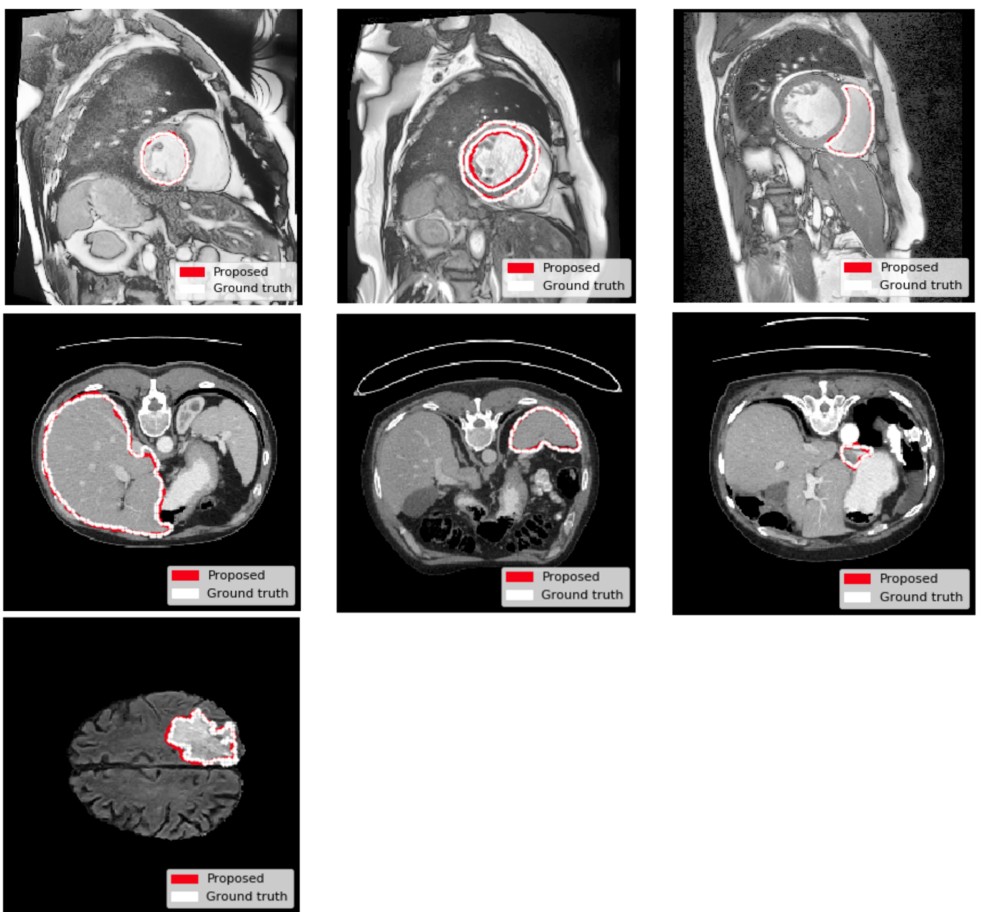

Figure A.1: Example results visualized for each dataset, starting from the top left and proceeding by row: endocardium, myocardium, right ventricle, liver, spleen, esophagus, and tumor.

performing method from the knowledge transfer approaches (PANet), the best performing method from the self- and semi- supervised approaches (CL), along with our proposed approach, the data augmentation approach, and the fully supervised network. We see that the WS approach behaves most similarly to the fully supervised networks. However, PANet actually is able to best generalize to the new test sets (relative to its original performance), possibly because of its inference procedure, which utilizes labeled images from the new test set.

