# OpenReview forum: "Cut out the annotator, keep the cutout: better segmentation with weak supervision"
_ICLR.cc/2021/Conference — ICLR 2021 Poster_

### Official Review · AnonReviewer2 · 2020-10-25
**Maybe insufficient methodological contribution; limited experimental evaluation and comparison with state-of-the-art**

**Rating:** 4
**Confidence:** 3

**Review:**

The authors employ ConvNets trained with a very small set of reference labeled images, and introduce a new weakly supervised model that focuses on certain areas of the image to fuse these weak sources (i.e. their approach introduces a conditional model that focuses on areas where the labeling functions disagree, more finely tracking the performance in such areas). I think the paper is interesting and well-presented, but there are several important aspects that, in my humble opinion, harm my overall rating of the paper:

- Not clear to me what is the main original contribution and novelty of this work. They use generic and simple labeling functions to obtain weak supervision and, then, they use a probabilistic model to refine the accuracy for each labeling function. As far as I knoe, both approaches (knowledge injection via weak segmentation algorithms, and probabilistic graphical models to model segmentation masks) are not novel, so what is the main methodological contribution and novelty of this work?

- Why the authors do not compare with other state-of-the-art few-shot medical image segmentation approaches (like [1])? In fact, [1] is already cited in the paper, but the experimental comparison does not include it. Why is that? Without a more extensive experimental comparison with prior methods is difficult to elucidate the actual empirical contribution of the proposed method.
[1] Ouyang, Cheng, et al. "Self-supervision with Superpixels: Training Few-Shot Medical Image Segmentation Without Annotation." ECCV 2020.

There are other details that should be clarified or improved in the paper:

- Are the authors employing one ("we use one-shot learning: we train a CNN from a single labeled data point for each LF.") or five ("five images are annotated by hand and used to seed LFs that weakly label the rest of the training set.") labeled images to weakly label their training data?

- The conclusions of the paper are too succint (just two lines and a half). I would encourage the authors to extend this section.

- What fully supervised network ("FS-all") do the authors use in their experiments?

---

> ### Author Response · Authors · 2020-11-22
> **Response to R2 (1/2)**
>
> We thank the reviewer for their comments, their time reviewing the paper, and their suggestions which helped us improve our submission. In a new thread to all reviewers, we address many of R2’s questions; below, we respond to each of R2’s comments individually.
>
>
> **Strengthened empirical evaluation**
>
> The reviewer asks, *“Why the authors do not compare with other state-of-the-art few-shot medical image segmentation approaches (like [1])?”* We compare against methods that have working, published implementations by the authors. Since the original submission deadline, the authors of [1] and [2] have released working codebases. We have updated our submission with results from these new baselines on our seven tasks in Section 4.2.1.
>
> Additional details of the strengthened empirical evaluation performed during this rebuttal period are provided in the thread to all reviewers. We include:
> * Results from the two new baselines [1,2]. In summary, we now compare against 5 recently-published baselines that represent varied approaches to FSL in addition to the lower-bound FS-5 and upper-bound FS-all. We find our proposed method exceeds the performance of all baselines on six of the seven segmentation targets. On average, we exceed the next-best baseline by 3.6 Dice points.
> * A new experiment on the impact of choosing different labeled reference points in Section 4.2.2.
> * Updated results achieved with our proposed method.
>
>
> **Clarification of the contributions**
>
> Our contributions are elaborated upon in the thread to all reviewers and have been concisely stated in the updated Introduction. Below we discuss R2's specific questions.
>
>
> * *“They use generic and simple labeling functions to obtain weak supervision and, then, they use a probabilistic model to refine the accuracy for each labeling function. As far as I know, both approaches (knowledge injection via weak segmentation algorithms, and probabilistic graphical models to model segmentation masks) are not novel so what is the main methodological contribution and novelty of this work?”*
>
> The novelty of our approach is in how we aggregate noisy labeling functions (LFs) to generate probabilistic training labels for segmentation models, which we define as WS. This approach to WS has seen widespread success in other domains [3,4] but has yet to be adapted to segmentation. There were two challenges in applying WS to segmentation use cases: defining LFs and defining a new PGM to aggregate LFs.
>
> 1. Labeling functions.
>
> Existing WS workflows require users to develop their own LFs for each task (example user-written LFs can be found at [11]). However, writing heuristics to define segmentation LFs is more difficult than for many NLP tasks, where WS has been largely used. Our solution is to propose simple FSL networks as LFs in WS. FSL models depart from the hand-written LF heuristics that are relied upon in existing WS tools.
>
> While FSL models alone have been studied widely, we propose to use FSL models as LFs in this new WS segmentation setting, which we show outperforms competitive FSL baselines.
>
> 2. Modeling.
>
> We agree that PGMs are a powerful tool for modeling images and have been actively researched for decades [5,6]. New PGMs to model segmentation masks have constituted significant contributions to computer vision [7,8]. Latent-variable PGMs have been developed to model the WS setting for classification use cases [9], where the goal is to aggregate multiple noisy LFs. Critical to WS is that the parameters of the PGM should be learned without any access to ground truth labels (if ground truth labels were available, WS wouldn’t be needed). However, existing WS PGMs do not work for segmentation (see details in thread addressed to all reviewers). The novelty of our contribution is proposing a new latent variable PGM, defined in Eqn. (2), which was designed specifically to aggregate multiple noisy segmentation masks.
>
> Our PGM estimates the conditional accuracy of each LF without relying on any ground truth segmentation masks---the parameters of the PGM are recovered from the noisy segmentation masks alone. The proposed PGM is unique:
> * It differs from past WS PGMs by introducing free parameters to focus on areas of contention, catching small areas of segmentation LF disagreement without introducing thousands of free parameters to model each edge in the graph.
> * It differs from many models used in PGM literature because the parameters of our model can be efficiently estimated without access to ground truth labels, making the model useful in the few-shot setting we investigate.
>
>
>
>
> Our novel pipeline is promising because:
> * We show it outperforms competitive FSL baselines on many segmentation targets.
> * It is orthogonal to future improvements in segmentation: as new FSL approaches are published, they can easily be integrated as LFs in our pipeline.

---

> > ### Author Response · Authors · 2020-11-22
> > **Response to R2 (2/2)**
> >
> > **Improved description of the methods**
> >
> > We have provided an improved description and a new figure of our weak labeling method in the thread addressed to all reviewers and in the updated Section 3.5, which should help clarify R2’s questions. Additionally, we directly answer each question below.
> >
> > * *Are the authors employing one or five labeled images to weakly label their training data?*
> >
> > We use five labeled images. Each labeled image is used to train a one-shot CNN, resulting in five total one-shot models (clarified in updated Figure 3 and Section 3.5).
> >
> > * *What fully supervised network ("FS-all") do the authors use in their experiments?*
> >
> > We use a 2-dimensional UNet [10] for FS-all, FS-5, and our proposed approach. For all baselines, we use the network provided in the corresponding authors’ published implementation of the baseline method. This information has been stated more clearly in Section 4.1 of the updated submission. Our code and networks will be public to enable replication of results.
> >
> >
> > Finally, we have expanded the conclusions of the paper, as recommended by R2, to include broader discussion of our proposed method and future work. We want to again thank R2 for their time reviewing the paper as well as for their suggestions to improve the submission. We hope that our clarifications of our contributions and the additional empirical evaluation address the reviewer’s primary concerns.
> >
> >
> >
> > [1] Cheng Ouyang, Carlo Biffi, Chen Chen, Turkay Kart, Huaqi Qiu, and Daniel Rueckert. Self-supervision with superpixels: Training few-shot medical image segmentation without annotation. ECCV, 2020.
> >
> > [2] Krishna Chaitanya, Ertunc Erdil, Neerav Karani, and Ender Konukoglu. Contrastive learning of global and local features for medical image segmentation with limited annotations. NeurIPS, 2020.
> >
> > [3] Stephen H. Bach, Daniel Rodriguez, Yintao Liu, Chong Luo, Haidong Shao, Cassandra Xia, Souvik Sen, Alex Ratner, Braden Hancock, Houman Alborzi, et al. Snorkel drybell: A case study in deploying weak supervision at industrial scale. International Conference on Management of Data, 2019.
> >
> > [4] Jared A Dunnmon, Alexander J Ratner, Khaled Saab, Nishith Khandwala, Matthew Markert, Hersh Sagreiya, Roger Goldman, Christopher Lee-Messer, Matthew P Lungren, Daniel L Rubin, et al. Cross-modal data programming enables rapid medical machine learning. Patterns, pp. 100019, 2020.
> >
> > [5] Daphne Koller and Nir Friedman. Probabilistic graphical models: principles and techniques. MIT press, 2009.
> > [6] Martin J Wainwright and Michael I Jordan. Graphical models, exponential families, and variational inference. Foundations and TrendsR in Machine Learning, 1(1-2):1–305, 2008.
> >
> > [7] Jamie Shotton, John Winn, Carsten Rother, and Antonio Criminisi. "Textonboost for image understanding: Multi-class object recognition and segmentation by jointly modeling texture, layout, and context." International journal of computer vision 81, no. 1 (2009): 2-23.
> >
> > [8] Philipp Krähenbühl, and Vladlen Koltun. "Efficient inference in fully connected crfs with gaussian edge potentials." In Advances in neural information processing systems, pp. 109-117. 2011.
> >
> > [9] Daniel Y Fu, Mayee F Chen, Frederic Sala, Sarah M Hooper, Kayvon Fatahalian, and Christopher Ré. Fast and three-rious: Speeding up weak supervision with triplet methods. ICML, 2020.
> >
> > [10] Olaf Ronneberger, Philipp Fischer, and Thomas Brox. "U-net: Convolutional networks for biomedical image segmentation." In International Conference on Medical image computing and computer-assisted intervention, 2015.
> >
> > [11] Alexander J Ratner, Christopher De Sa, Sen Wu, Daniel Selsam, Christopher Ré. Data programming: Creating large training sets, quickly. NeurIPS, 2016.

---

### Official Review · AnonReviewer1 · 2020-10-27
**The authors make a clear explanation of the problem and rationale behind their approach, but a few experimental questions should be addressed concerning the implementation of the models and needs some additional results.**

**Rating:** 6
**Confidence:** 4

**Review:**

CUT OUT THE ANNOTATOR, KEEP THE CUTOUT: BETTER SEGMENTATION WITH WEAK SUPERVISION
The authors make a clear explanation of the problem and rationale behind their approach, but a few experimental questions should be addressed concerning the implementation of the models and needs some additional results.

*Review
The paper is well written with significant time spent setting up a clear explanation of the background information and challenges.
The authors also adequately discuss alternate approaches and other methods currently used to address this current problem.
The raw numeric results of the proposed method are very compelling.

The authors make a good case for the significance of this work if it is able to surpass current methods in this space.
There are several questions that are still left un-addressed:

-Because the size of a neural network impacts the quality of the segmentation, it is important to know the size of the U-Net that they used for each model. The proposed method uses several CNNs in combination. So depending on the size of networks used in the proposed method (compared to the network size of the other chosen methods), the results might be biased based solely on the size of the networks used.

-The authors explain that the stacked networks will not add computational time due to the fact that they are each trained on only 1 image. This might be small since they were only given 5 training images, but this time will scale with more training data. Also, when stacked networks are applied to large datasets at test time, the additional computational times will add up. It would be nice to see a numeric comparison of computational times.

-One problem of learning from small datasets is guaranteeing balance and the quality of the training images provided (Changing out even a single image can impact model training). How variable are the results on what training images are provided? Authors state that they trained the models three times with different seeds, but it is assumed they used the same training images in every iteration.

-Authors state that random image augmentations were used in all models except FS-5 (unstated what was used for FS-all). Since image augmentation is important to apply to test set and generalization in general, could authors provide the rationale as to why baseline models weren’t trained with image augmentation?

- In Table 1 three trained models were averaged. It would be nice to also show deviations alongside the means so we can see the consistency of each approach.

- In Figure 4. Scaling of the axis could be misleading. Keep colors consistent for the proposed method.

---

> ### Author Response · Authors · 2020-11-22
> **Response to R1 (1/2)**
>
> We thank R1 for their time reviewing the manuscript and for their suggestions on how to improve our submission. We were glad to receive the reviewer’s positive comments regarding the problem set-up and empirical results. In a new thread addressed to all reviewers, we have provided clarifications to our method’s workflow and descriptions of new experiments. Below, we address each of R1’s comments individually.
>
>
> **New empirical evaluation**
>
> R1 brings up the great point that few-shot models can be sensitive to which labeled training images are provided. We thank the reviewer for this point, as it is an interesting and important experiment. In response, we have added new empirical evaluations where we train our proposed method with 3 new sets of labeled images. Details of this new experiment can be found in the thread addressed to all reviewers and in Section 4.2.2 of the updated paper. In summary, we find good agreement between selection of different labeled training images, with five of the seven datasets having a standard deviation less than 1.5 Dice points over four sets of different labeled training images.
>
> In the common thread to all reviewers, we also describe the results of two new baselines we have compared against in the rebuttal period.
>
>
> **Clarifying our method’s workflow**
>
> Multiple of R1’s questions have to do with the size and training time of the CNNs used in this work. We believe many of these concerns will be resolved by clarifying our workflow, which we have done in the new thread addressed to all reviewers and in Section 3.5 of our updated submission. We directly answer R1s questions here:
>
>
> * *“Because the size of a neural network impacts the quality of the segmentation, it is important to know the size of the U-Net that they used for each model. The proposed method uses several CNNs in combination. So depending on the size of networks used in the proposed method (compared to the network size of the other chosen methods), the results might be biased based solely on the size of the networks used.”*
>
> Each of the LF CNNs in Step 1 and the end model CNN in Step 3 is a 2D UNet, as described in the original UNet paper [1].
>
> We clarify that the CNNs are used in a stage-wise manner---they are not stacked in the final evaluation of the network. The labeling function CNNs in Step 1 of our workflow are used to weakly label all *training* images. The end model CNN in Step 3 produces labels for all *testing* images. In other words, the final network output from our workflow is just a single CNN, and the test images are not processed by the labeling functions in Step 1, which are solely used to weakly label the training data.
>
> As a result, there is no difference in network capacity of the final segmentation network. Our final segmentation CNN---which produces the segmentation masks for all test images---is just a single 2D UNet, which is similar to the networks used in the baselines we compare against [2,3,4,5,6]. The similar network capacities are why we claim that our performance boosts come from our approach to leveraging unlabeled and few labeled datapoints.
>
> * *“The authors explain that the stacked networks will not add computational time due to the fact that they are each trained on only 1 image. This might be small since they were only given 5 training images, but this time will scale with more training data.”*
>
> We clarify that while we only used 5 labeled training images, we had many more unlabeled images. Leveraging these unlabeled images to train a supervised network is how we are able to achieve such strong results. We anticipate our methods being used in circumstances when the ratio of unlabeled to labeled data is large. Training time of the labeling functions will not increase with more unlabeled images, since the labeling functions are only trained using the small number of labeled training images.
>
>
>
> * *“Also, when stacked networks are applied to large datasets at test time, the additional computational times will add up. It would be nice to see a numeric comparison of computational times.”*
>
>
> As described above, there are no stacked networks used at test time. The labeling function CNNs are solely used to weakly label the training data. The final segmentation network is just a single CNN---the final architecture is not bound to the labeling functions. Thus, computation time at inference will be equivalent to any method that uses a 2D UNet.
>
> We believe the modularity of our system is an important distinction, so we appreciate R1’s concerns and have updated the description of our workflow in Section 3.5 to clarify this for future readers.

---

> > ### Author Response · Authors · 2020-11-22
> > **Response to R1 (2/2)**
> >
> > **Smaller clarifications and adjustments**
> >
> > * FS-all was trained with the same data augmentation used in our proposed method. The only network for which data augmentations were not used was FS-5, since FS-5 is intended to be a lower-bound on performance.
> > * We have added standard deviations to the results in all Tables.
> > * We have changed the colors in Figure 5 (previously Figure 4).
> >
> >
> > We again thank the reviewer for their time and suggestions for improving our submission.
> >
> >
> > [1] Olaf Ronneberger, Philipp Fischer, and Thomas Brox. "U-net: Convolutional networks for biomedical image segmentation." In International Conference on Medical image computing and computer-assisted intervention, 2015.
> >
> > [2] Cheng Ouyang, Carlo Biffi, Chen Chen, Turkay Kart, Huaqi Qiu, and Daniel Rueckert. Self-supervision with superpixels: Training few-shot medical image segmentation without annotation. ECCV, 2020.
> >
> > [3] Krishna Chaitanya, Ertunc Erdil, Neerav Karani, and Ender Konukoglu. Contrastive learning of global and local features for medical image segmentation with limited annotations. NeurIPS, 2020
> >
> > [4] Kaixin Wang, Jun Hao Liew, Yingtian Zou, Daquan Zhou, and Jiashi Feng. Panet: Few-shot image semantic segmentation with prototype alignment. ICCV, 2019.
> >
> > [5] Abhijit Guha Roy, Shayan Siddiqui, Sebastian Polsterl, Nassir Navab, and Christian Wachinger. ‘Squeeze & excite’ guided few-shot segmentation of volumetric images. Medical image analysis, 2020.
> >
> > [6] Krishna Chaitanya, Neerav Karani, Christian F Baumgartner, Anton Becker, Olivio Donati, and Ender Konukoglu. Semi-supervised task-driven data augmentation for medical image segmentation. International Conference on Information Processing in Medical Imaging, 2019.

---

### Official Review · AnonReviewer4 · 2020-10-27
**Interesting approach to medical image segmentation with limited labelled data, evaluation could be stronger**

**Rating:** 7
**Confidence:** 4

**Review:**

The paper addresses the problem of performing medical image segmentation in the limited label scenario, i.e. the case where there is only a small number of manual expert segmentations available.

The proposed approach to address this challenge uses a combination of two approaches: First, multiple labelling functions are learned by training CNNs with a single labelled example. The resulting labelling functions are then combined using a label fusion approach. This label fusion approach is based on a conditional model that focuses on areas of contention. To the best of my knowledge, the proposed approach is novel.  I found the methodology interesting and well motivated. The formulation and writing is clear. The main weakness to me is the evaluation. First, the selected baselines are not always very strong. For example, the best performing method for segmentation of ACDC (cardiac MRI) reported in Bernard et al., 2018 achieves significantly higher results than the baseline segmentation reported (”FS-all”).

The authors mention that, depending on task difficulty, more sophisticated limited label learning models could be used as labelling functions. It would be good to see some discussion of this.

---

> ### Author Response · Authors · 2020-11-22
> **Response to R4**
>
> We thank the reviewer for their positive comments, their detailed read of the paper, and their suggestions for improving the paper. Below we address each of the reviewer’s comments.
>
> * *The main weakness to me is the evaluation.*
>
> We describe new experiments to strengthen our evaluation in a separate thread addressed to all reviewers, which we ask R4 to please read if interested in more details. In summary, we include: (1) Two new few-shot methods that have been tested on our datasets [1,2]. (2) A new experiment we perform on the impact of which few images are labeled. (3) Improved results achieved with our proposed method.
>
>
> * *First, the selected baselines are not always very strong. For example, the best performing method for segmentation of ACDC (cardiac MRI) reported in Bernard et al., 2018 achieves significantly higher results than the baseline segmentation reported (”FS-all”).*
>
> We thank the reviewer for pointing this out, and we have updated Section 4.1 to address this concern for future readers. There are many reasons why FS-all in our paper is lower than fully supervised performance reported elsewhere, such as the reference R4 includes. At a high level, the aim of our paper was to show how well we could utilize unlabeled data and a small set of labeled data via our proposed method, and to compare our proposed method to existing few-shot baselines. As a result, we did not implement complex architectures or training schedules for our end networks. Instead, we used the same simple 2D UNet as an end model for all experiments, which allowed us to establish fair comparisons between (1) our proposed method and baseline few-shot approaches, as well as (2) our proposed method and FS-all.
>
> Had we significantly increased the capacity of the end model in our proposed method and FS-all, the performance gains of our proposed pipeline could be attributed to this more complex end model and not our particular approach to weakly labeling data. By instead using a simple 2D UNet, which is similar to the networks used in the baselines we compare against [1,2,3,4,5], we can claim that our performance gains are due to our method of using limited labeled data.
>
> Reasons that contribute to the performance differences between our FS-all and [6] include their use of: a 3D network in addition to a 2D network; additional images from other phases of the cardiac cycle; an ensemble of end networks; and additional connections in the 2D and 3D UNet architectures. For future use, the network of [6] could be easily substituted into Step 3 of our proposed method to improve the raw performance metrics.
>
>
> * *The authors mention that, depending on task difficulty, more sophisticated limited label learning models could be used as labelling functions. It would be good to see some discussion of this.*
>
> We appreciate this suggestion, and we have updated our conclusion to include more substantial discussion of how the labeling functions could be improved. In the new conclusion, we discuss how unsupervised pre-training could be used to initialize the one-shot networks used in the labeling functions, such as the pretraining methods proposed in [1,2]. These unsupervised pre-training tasks would leverage the unlabeled data to help the one-shot networks learn more robust representations of the images prior to being fine tuned with a single labeled image. In fact, any of the baselines we compare against in this work [1,2,3,4,5] could be used as a labeling function in our pipeline. Because our system is modular, labeling functions can easily be substituted as new few-shot segmentation approaches are published.
>
>
> We again thank the reviewer for their positive comments and suggestions to further improve our work.
>
>
>
> [1] Cheng Ouyang, Carlo Biffi, Chen Chen, Turkay Kart, Huaqi Qiu, and Daniel Rueckert. Self-supervision with superpixels: Training few-shot medical image segmentation without annotation. ECCV, 2020.
>
> [2] Krishna Chaitanya, Ertunc Erdil, Neerav Karani, and Ender Konukoglu. Contrastive learning of global and local features for medical image segmentation with limited annotations. NeurIPS, 2020
>
> [3] Kaixin Wang, Jun Hao Liew, Yingtian Zou, Daquan Zhou, and Jiashi Feng. Panet: Few-shot image semantic segmentation with prototype alignment. ICCV, 2019.
>
> [4] Abhijit Guha Roy, Shayan Siddiqui, Sebastian Polsterl, Nassir Navab, and Christian Wachinger. ‘Squeeze & excite’ guided few-shot segmentation of volumetric images. Medical image analysis, 2020.
>
> [5] Krishna Chaitanya, Neerav Karani, Christian F Baumgartner, Anton Becker, Olivio Donati, and Ender Konukoglu. Semi-supervised task-driven data augmentation for medical image segmentation. International Conference on Information Processing in Medical Imaging, 2019.
>
> [6] F. Isensee, P. Jaeger, P. M. Full, I. Wolf, S. Engelhardt and K. H. Maier-Hein, "Automatic cardiac disease assessment on cine-MRI via time-series segmentation and domain specific features." STACOM-MICCAI, 2017.

---

### Official Review · AnonReviewer3 · 2020-10-28
**This paper provides a weakly supervised method for medical image segmentation. However, it has several weaknesses need to be addressed in the future.**

**Rating:** 5
**Confidence:** 4

**Review:**

This paper presents weakly supervised framework for image segmentation tasks with limited annotated data. It first builds several labeling functions with limited annotation and then uses probability graph to fuse the labels.  Last, the final output will be generated via CNN network.  There are two key challenges in such setting. First, how to build the labeling functions. Second, how to measure the accuracies from the labeling functions.

However, these two challenges are not well described in this paper. First, the labeling functions are built on prior knowledges such as physical models, clustering and pretrained network.  Especially , for the pretrained network, it is not fair in few shot learning setting.  Second, the author claims she/he uses one image to train the LFs, since applying one image to train the network is difficult, it is very confusing on how to train the network. Third,  the author uses 60% data for training, it is not clear how to differentiate the proposed weakly supervised training with traditional supervised training (it is not uses true label?but the paper not describe it).  Morever, since FS-5 has reasonable good performance, the author should also compares with FS-60%.

Since current paper has lots parts need to be cleared, the current draft can not be accepted.


-----------------------------------------------------------------------------------------------------------------------------------
The author answers to reviewers' questions clearly with supportive details.  In addition,  the newer version has added useful new baselines and results. I have increased my score as all my concerns get cleared.

Overall,  I think  this is an interesting paper and should be encouraged. But I still has some concerns as the application is very limited on medical images. The author has compared all the baselines and shown that the generalization capability for the proposed method is similar to fully supervised method, which seems not doing well.  I would be more convinced if the proposed can be tested on few shot benchmark which are natural images.

---

> ### Author Response · Authors · 2020-11-22
> **Response to R3**
>
> We thank R3 for their time in reviewing the paper and providing feedback. We provide information to all reviewers in a new thread, including (1) additional experimental results, (2) discussion of our contributions, and (3) an updated description detailing our method. We ask R3 to please read the thread addressed to all reviewers, as it is relevant to the concerns raised in R3’s review. Below, we provide additional detail to address R3's specific questions.
>
> We believe that there are several misunderstandings in this review that are important to address as they do not reflect the method we proposed. We hope that the discussion helps clarify, and that the adjustments we made to the paper prevent these concerns for future readers.
>
> * *“However, these two challenges are not well described in this paper.”*
>
> We have discussed these challenges more extensively in new thread to all reviewers, and we have reformatted Section 3 to make these challenges more clear.
>
> * *“First, the labeling functions are built on prior knowledges such as physical models, clustering and pretrained network. Especially , for the pretrained network, it is not fair in few shot learning setting.”*
>
> We clarify that our LFs are not built on prior knowledge. Our LFs are CNNs trained with a single labeled image---there are no pretrained networks or physical models used to achieve the results presented in Section 4. Since each of our LFs is a one-shot CNN, it is fair to compare our proposed approach to baseline few-shot learning methods: all methods have access to the same five labeled images, and no pretrained networks are used. To clarify this, we have updated the description and figure of our proposed workflow in the thread addressed to all reviewers and in Section 3.5 of the updated submission.
>
> We included alternative LFs in Section 3.2 of the original submission to describe other potential segmentation LFs---such as physical models and pretrained networks---that could be useful in future research. Because our method is performed in stages, our PGM can aggregate noisy labels from any LF, not just the one-shot CNNs used in the submission. We view this flexibility as an advantage of our proposed approach. However, we have seen that the discussion about alternative LFs has generated questions for multiple reviewers. In response, we have moved the discussion to the Appendix.
>
>
> * *“Second, the author claims she/he uses one image to train the LFs, since applying one image to train the network is difficult, it is very confusing on how to train the network.”*
>
> The reviewer asks about how each one-shot CNN was trained. We train a CNN using a single image volume and extensive data augmentation (elastic transforms, contrast jitters, and affine transforms), which has also been shown in previous works as capable of generating segmentation masks (an example here [1]). We note that the outputs of these one-shot LFs are indeed noisy---we reduce this noise using our PGM, and the resulting weak labels are sufficient for training an end model. We have updated Section 4.1 to state this more clearly.
>
>
> * *“Third, the author uses 60% data for training, it is not clear how to differentiate the proposed weakly supervised training with traditional supervised training (it is not uses true label?but the paper not describe it). Morever, since FS-5 has reasonable good performance, the author should also compares with FS-60%.”*
>
> The reviewer states that we use 60% of the dataset as labeled training data, however this is not true: we use only five labeled images, the rest of our training set is initially unlabeled.
>
> We believe this concern is due to our statement, “each dataset is split into 60% training, 20% validation, and 20% testing,” which appears in Section 4. This means that 60% of all images in the dataset are used during the training stage. However, only five of these training images are labeled. The remaining unlabeled images in the training set are weakly labeled via labeling functions and a probabilistic graphical model. We have made this distinction more clear in the updated submission in Section 3.5 and in the new Figure 3. Each of the baselines we compare against (FS-5 as well as all previously proposed approaches) also use the same five labeled images. We believe this is an important distinction to clarify, as our results in Section 4 are achieved with many fewer labeled images than thought by the reviewer.
>
>
> We hope the updated submission resolves the concerns R3 has about the number of labeled images used in the proposed approach, as well as how these labeled images are utilized in labeling functions. We thank you again for your time and feedback in helping us to improve our submission.
>
>
>
> [1] Amy Zhao, Guha Balakrishnan, Fredo Durand, John V. Guttag, and Adrian V. Dalca. "Data augmentation using learned transformations for one-shot medical image segmentation." In Proceedings of the IEEE conference on computer vision and pattern recognition, 2019.

---

### Official Review · AnonReviewer5 · 2020-11-07
**Interesting application; Limited technical contribution, unclear writing, and unconvincing experiment results.**

**Rating:** 6
**Confidence:** 2

**Review:**

### Summary
This paper studies few-shot segmentation for medical images where labeled data is hard to obtain.  It proposed a stage-wise pipeline where pseudo labels are first proposed by noisy learning functions (LFs), then aggregated through a PGM, and finally used to train a segmentation model. The authors argue the two biggest challenges are "injecting knowledge for segmentation model" and "measuring accuracies of pseudo labels", and propose two modules to address them. The experiments have shown that the proposed method outperforming several few-shot learning baselines.

### Pros
The studied application is indeed very important and requires few-shot learning techniques. The 2nd challenge ("measuring accuracies of pseudo labels") is very insightful and indeed a huge problem for many methods. The technical solution of using PGMs is interesting and should be encouraged.  The experimental results are promising and encouraging.

### Cons
1. The abbreviations are abused, especially about FSL, LLL, and WS. As a reviewer from vision community, there is a common agreement of the definition of FSL and WS. LLL is rarely used and usually represents a very broad concept rather then anything specific. This paper has different interpretations of these concepts, which is fine, but should make it really clear what they stands for and what are the differences between them. In my opinion, this paper is simply a FSL setting and there are no need to introduce other two terms.
2. The notations are also not super clear to me. In Sec.3.1, it's said $X_i \in \mathbb{R}^{I,J,K,T}$ is a tensor of $P$ pixels. However, $P$ doesn't go into any places of the notation. Moreover, what are $I,J,K,T$? They are never properly introduced. Also,  the superscript and subscript in $X_{i=1}^{i=n}$, $Y_{i=1}^{i=n}$ are just self-contradictory.
3. In Sec.3.2 four LFs are discussed (Physical models, pre-training networks, ...). Are these methods ever used in the proposed framework, or compared to as baselines in experiments? Also, the reference of these methods are missing.
4. It seems like I miss something. There seem to be only one label function:  a few-shot image classifier. What are the multiple LFs?
5. The design choices are not properly ablated. It's unclear which part contributes the most.
6. The FSL baselines are very limited: only PANet. In fact there are many orthogonal solutions of FSL. It's preferable multiple different baselines are compared  for this relatively new task.
7. Tab.2 results are obtained by directly comparing accuracy numbers for two different dataset, which doesn't make sense to me as they are not comparable. A better way to evaluate generalization performance is to compare "test a model pretrained using A on B" and "test a model trained on B".

I have some doubts about the evaluation:
1. What's the reason for using Dice as evaluation metric rather than standard IoU, Pixel-Acc metrics? DICE is not popular for any segmentation work, and from the formulation it seems to be a worse metric than IoU (union area summed twice in the denominator).

I have some doubts about the motivation:
1. Why "Injecting knowledge via LFs" is a big challenge? Isn't it the standard pseudo-label or knowledge distillation algorithm? Multiple semi-supervised/weakly-supervised segmentation papers have studied this idea. To name a few:
*. Papandreou and Chen etal., Weakly- and Semi-Supervised Learning of a Deep Convolutional Network for Semantic Image Segmentation. ICCV 2015
*. Wei etal., STC: A simple to complex framework for weakly-supervised semantic segmentation. PAMI

### Misc
1. Since all the experiments have been runned 3 times and the labeled samples are few (5), it's worth reporting the variance/std together with the mean.  Some results in Tab.1 are very close and hard to tell which method is better.
2. Sec.3.2 `` "LLL labeling functions" writs "one-shot learning". However, in experiments it seems like 5 labeled images are used. Better to be rigorous.
3. What's the dimension of $\theta_i$ in Eq.(1)?
4. Fig.4 can be improved to be more consistent via using the same color for proposed method.
5. I would change Fig.1 example to use medical images.

### Questions
1. Have the authors tried to sample different labeled images? Will the results be different? Compared to using random seeds, I would report mean results over different samples.

********************************
The rebuttal is very detailed and contains many useful new baselines and results. The authors answer to reviewers' questions carefully and with great supportive details. I appreciate the efforts and many of my questions are answered, therefore I have increased my score.

I'm not  an expert from medical image field. So it's a little bit hard for me to evaluate the significance of the reported results and proposed methods. However, I think this paper is well-motivated and current version has greatly improved over the first draft.  For the machine learning community, it's an interesting and important application and thus should be encouraged.

---

> ### Author Response · Authors · 2020-11-22
> **Response to R5 (1/3)**
>
> We want to thank R5 for their detailed comments and time reviewing the paper. We are glad that R5 found the application interesting and values the technical solution of using the new PGM to resolve accuracies of noisy labeling sources. We found the reviewer’s questions and suggestions particularly helpful in improving our manuscript. We have included a response to all reviewers in a new thread and address each of R5’s comments below.
>
> **Strengthened empirical evaluation**
>
> We describe new empirical evaluation in the thread to all reviewers, including:
> * Results from two new FSL baselines. In summary, we now compare against 5 recently-published baselines [5,6,7,8,9] that represent varied approaches to FSL in addition to the lower-bound FS-5 and upper-bound FS-all. We find our proposed method exceeds the performance of all baselines on six of the seven segmentation targets. On average, we exceed the next-best baseline by 3.6 Dice points.
> * Results from a new experiment requested by R5 about how different labeled training points impact results. We appreciate this comment from R5, as it is an important experiment. In summary, we find good agreement between runs with different labeled images, with five of the seven segmentation targets having standard deviations <=1.5 Dice points between runs with different sets of labeled images.
> * Improved results with our proposed method.
>
>
> **Clarifying and updating terminology**
>
> There is an ambiguity of terms that partially results from “weak supervision” (WS) being overloaded in literature and we appreciate R5’s suggestions for addressing this ambiguity. We have updated the paper such that the primary terms used are WS and few-shot learning (FSL).
>
> * *Weak supervision*. We define WS as an approach to generating and aggregating multiple noisy labeling sources to compute probabilistic training labels. These training labels can then be used to train any downstream network. This problem setup is developed in prior work [1,2] and has been deployed across domains [3,4]. R5 suggests that WS does not need to be used as a term in the manuscript. However, because we build upon WS principles as developed in previous work, we believe it is an important term to include. We have included a new Section 3.1 to clarify this term.
> * *Few-shot learning*. We have broadened our definition of FSL to include all methods of training networks with few labeled examples.
>
>
> **Clarifying novelty and contributions**
>
> We have discussed our primary contributions in the thread addressed to all reviewers and added concise statements of contribution in our updated Introduction. Below we address R5’s questions directly.
>
>
> *Why is “injecting knowledge via LFs” a challenge? Isn't it the standard pseudo-label or knowledge distillation algorithm? Multiple semi-supervised/weakly-supervised segmentation papers have studied this idea. To name a few: ...*
>
> Supervising networks with noisy labels is indeed central to many active areas of ML research—knowledge distillation, pseudo-labels, and using bounding boxes or image-level labels for semantic segmentation are a few examples. The difference of our contribution is that we propose a new approach to aggregating the outputs of multiple noisy segmentation sources to generate training labels for downstream networks. This WS approach has been successful in other domains [3,4] and we hypothesized it could improve few-shot segmentation. To develop this WS segmentation pipeline, we had to solve two challenges that make using existing WS tools difficult for segmentation: defining LFs and defining a new PGM. We discuss the first challenge here, which R5 has asked about.
>
> * Existing WS pipelines rely on LFs developed by users. Writing LFs to express segmentation heuristics is more difficult than for NLP and classification tasks, where LFs have been widely deployed. We show that using FSL models as LFs results in high performance in a WS pipeline and does not require the user to write their own LFs.
> * Although FSL and various methods of generating pseudo-labels have been studied before, we propose to use FSL models as LFs in our WS segmentation setting, overcoming one of the challenges associated with extending existing WS tools to segmentation.
> * As a minor point, the two references provided by R5 require inputs that we do not have access to in the datasets used here: pre-trained networks and image-level labels or bounding boxes for all training images. For tasks where this signal is available, the papers referenced by R5 could be used as LFs in our pipeline.
>
> Our proposed approach is promising because:
> * We show it outperforms competitive FSL baselines on many segmentation tasks.
> * It is orthogonal to future improvements in segmentation: as new FSL approaches are published, they can easily be integrated as LFs into our pipeline.

---

> > ### Author Response · Authors · 2020-11-22
> > **Response to R5 (2/3)**
> >
> > **Improved description of methods**
> >
> > We have improved the description of our workflow in the new Section 3.5, which is also stated in the thread addressed to all reviewers, to address R5’s questions for future readers. We directly answer R5’s questions below.
> >
> > * *What are the multiple labeling functions?*
> >
> > Each LF is a CNN trained with one image volume. Because we have five labeled image volumes, we have a total of five LFs (clarified in the new Figure 3 of the updated submission).
> >
> > * *Are the LFs proposed in Section 3.2 ever used in the proposed framework?*
> >
> > We included alternative LFs in Section 3.2 to suggest other potential segmentation LFs that readers could use for other tasks---these alternative LFs may provide additional signal to the PGM for other tasks (e.g. pretrained networks will likely provide useful signal when available). This underlines a strength of our system: the PGM can compute the conditional accuracies of any set of LFs, not just the one-shot CNNs we used in our work. Using our PGM, it is easy to include many different sources of signal in the pipeline.
> >
> > While we experimented with the other LFs described in the original Section 3.2, they are not as general as the FSL models that we use in the submission, which can be applied to many tasks (e.g., pretrained networks are often not available, thresholding or clustering are too simple for many tasks). We have moved this discussion to the Appendix to prevent similar confusion for future readers.
> >
> >
> > **Remaining questions and adjustments made in response to R5’s comments**
> >
> > * *Why is the DICE score used instead of IoU and pixel-acc metrics?* We use the DICE score because it is the performance metric reported in four of the five baselines we compare against [5,6,8,9] and is the common metric in the leaderboards for the public datasets we use. DICE is a common metric in medical image segmentation. By reporting DICE, we enable comparison of our results to similar published papers and baselines.
> > * *“Tab. 2 results are obtained by directly comparing accuracy numbers for two different dataset...a better way to evaluate generalization performance is to compare ‘test a model pretrained using A on B’ and ‘test a model trained on B.’”* We appreciate this point. Unfortunately, because large, labeled segmentation datasets are not common, there were two obstacles stopping us from performing the requested evaluation. (1) The new datasets do not contain labels for all classes. Because the knowledge transfer networks require labels from other classes to be present during training, we could not train a network using multiple of our baselines on the new datasets. (2) The new CHAOS dataset (n=20) is too small to partition into training, validation, and testing splits. These obstacles are why we reported results relative to performance on the original datasets. In the updated submission this experiment has been moved to the Appendix, as other additions from this rebuttal were prioritized for space.
> > * Ablations have been added to the Appendix.
> > * Notation in Section 3.1 has been adjusted.
> > * Standard deviations have been included in the results.
> > * Dimensions for all canonical parameters in Eqn. (1) and Eqn. (2) have been added.
> > * Figure 5 (previously Figure 4) colors have been adjusted.
> > * We tried changing Figure 1 to a medical image. However we found this made the figure less intuitive for readers unfamiliar with medical image segmentation targets, whereas the bird segmentation is easy for readers from different backgrounds to follow (e.g., what the ground truth segmentation should be, and thus what was noisy about the LF outputs).
> >
> >
> > We hope the additional experimental results, updated description of the workflow and terminology, and discussion of contributions address your concerns. We thank you again for your time reviewing the paper and insightful comments, which have helped us strengthen our manuscript.

---

> > > ### Author Response · Authors · 2020-11-22
> > > **Response to R5 (3/3)**
> > >
> > > [1] Alexander J Ratner, Stephen H Bach, Henry Ehrenberg, Jason Fries, Sen Wu, and Christopher Ré. Snorkel: Rapid training data creation with weak supervision. VLDB, 2018.
> > >
> > > [2] Alexander J Ratner, Christopher De Sa, Sen Wu, Daniel Selsam, Christopher Ré. Data programming: Creating large training sets, quickly. NeurIPS, 2016.
> > >
> > > [3] Stephen H. Bach, Daniel Rodriguez, Yintao Liu, Chong Luo, Haidong Shao, Cassandra Xia, Souvik Sen, Alex Ratner, Braden Hancock, Houman Alborzi, et al. Snorkel drybell: A case study in deploying weak supervision at industrial scale. International Conference on Management of Data, 2019.
> > >
> > > [4] Jared A Dunnmon, Alexander J Ratner, Khaled Saab, Nishith Khandwala, Matthew Markert, Hersh Sagreiya, Roger Goldman, Christopher Lee-Messer, Matthew P Lungren, Daniel L Rubin, et al. Cross-modal data programming enables rapid medical machine learning. Patterns, pp. 100019, 2020.
> > >
> > > [5] Cheng Ouyang, Carlo Biffi, Chen Chen, Turkay Kart, Huaqi Qiu, and Daniel Rueckert. Self-supervision with superpixels: Training few-shot medical image segmentation without annotation. ECCV, 2020.
> > >
> > > [6] Krishna Chaitanya, Ertunc Erdil, Neerav Karani, and Ender Konukoglu. Contrastive learning of global and local features for medical image segmentation with limited annotations. NeurIPS, 2020.
> > >
> > > [7] Kaixin Wang, Jun Hao Liew, Yingtian Zou, Daquan Zhou, and Jiashi Feng. Panet: Few-shot image semantic segmentation with prototype alignment. ICCV, 2019.
> > >
> > > [8] Abhijit Guha Roy, Shayan Siddiqui, Sebastian Polsterl, Nassir Navab, and Christian Wachinger. ‘Squeeze & excite’ guided few-shot segmentation of volumetric images. Medical image analysis, 2020.
> > >
> > > [9] Krishna Chaitanya, Neerav Karani, Christian F Baumgartner, Anton Becker, Olivio Donati, and Ender Konukoglu. Semi-supervised task-driven data augmentation for medical image segmentation. International Conference on Information Processing in Medical Imaging, 2019.

---

### Author Response · Authors · 2020-11-22
**Response to all reviewers (1/4)**

We first thank all reviewers for their time reviewing the paper. We appreciate the many detailed suggestions which have helped us to improve our submission. In response, we are providing the following:
1. This thread, which presents new evaluation and clarifies questions asked by multiple reviewers.
2. Point-by-point responses in each reviewer’s thread.
3. An updated submission incorporating feedback from all reviewers.

Many reviewers recognized the importance of the problem, the advantages of the proposed approach, and the promise in the empirical results, and we appreciate these positive comments. Below, there is a new comment for each of the three primary areas we have strengthened our paper: strengthened empirical evaluation, clarification of novelty and contributions, and improved description of methods.

---

> ### Author Response · Authors · 2020-11-22
> **Response to all reviewers (2/4): Strengthened empirical evaluation: new results from two recently-published baselines, reviewer-recommended experiments, and improved results with proposed method.**
>
> In response to the reviewers’ comments, we have run new experiments to include in our empirical evaluation (detailed below and included in the updated Section 4).
>
> **Additional baselines**
> Since the original submission deadline, two new relevant approaches to few-shot medical image segmentation have released working codebases: a self-supervised approach [1] and a contrastive learning approach [2]. We have included new evaluations of these two approaches in Section 4, specifically Table 1.
>
> In summary, we now compare against five FSL approaches: a self-supervised approach (“SSL-ALPNet”, [1]), a contrastive learning approach (“CL”, [2]), PANet [3], Squeeze and Excite (“S&E”, [4]), a GAN data augmentation method (“GAN DA”, [5]), and the lower-bound FS-5 and upper-bound FS-all. These baselines cover a wide range of FSL segmentation methods and have been recently published in ML, CV, and medical imaging venues. Through our empirical evaluation, we show that the performance of our proposed method exceeds all others on six of the seven segmentation tasks.
>
>
> **Impact of labeled reference images**
> R5 and R1 asked how results change depending on the selection of the five manually labeled images. We’re glad to receive this suggestion, as it’s an interesting experiment. The original five labeled reference points were randomly selected and used in all baselines and our proposed approach. In the updated submission, we have run our proposed pipeline with three new sets of labeled images to evaluate how performance changes; all sets are mutually exclusive. The results of this experiment are shown in Table 2 of Section 4.2.2 in our updated submission. In general, we see good agreement across different randomly-selected labeled images. For all datasets except the right ventricle and the esophagus, we see standard deviations less than 1.5 Dice points.
>
> We note that all few-shot approaches will be impacted by the selection of the few labeled datapoints, though many of the previously-published baselines that we compare against do not discuss the impact of which images are labeled. While we do not have the compute to run all baselines with new labeled images before the rebuttal deadline, we will continue running this experiment in case of the camera-ready deadline.
>
>
> **Improved results of proposed method**
> Finally, we have added a step to the end of our proposed pipeline, which has improved results of our method (updated results shown in Tables mentioned above). After training the end model in Step 3 with the weak labels, we fine-tune the network with the five manually labeled images.

---

> > ### Author Response · Authors · 2020-11-22
> > **Response to all reviewers (3/4): Clarification of novelty and contributions.**
> >
> > We include succinct statements of contribution in the updated Introduction and have reformatted Section 3 to make these contributions more clear. Below, we discuss these contributions.
> >
> > As background, we build our approach off of weak supervision (WS), which we note is an overloaded term in literature. We refer to WS as the class of approaches in which more than one noisy labeling source (or labeling function, LF) can be defined and aggregated into probabilistic training labels without access to ground truth labels [6,7]. We have updated our submission to clarify this distinction in Section 3.1. WS has produced state-of-the-art results across many domains [8,9,10]. We hypothesized that we could use WS to achieve state-of-the-art results in segmentation with limited labels, but to do so we had to overcome two challenges: defining LFs for segmentation and developing a new PGM to aggregate LF outputs.
> >
> > **Contribution 1: A novel WS approach to training segmentation networks with few manually-annotated images. Our proposed approach exceeds the Dice score of the next-best baseline by a mean 4.7% and gets within a mean 5.4% of a network trained with all manually-annotated images.**
> >
> > Existing WS tools require the user to define their own LFs (examples in [7]). However, writing heuristics to define image segmentation LFs is more difficult than for NLP and classification tasks, where WS has been widely deployed. We solve this problem by showing that simple few-shot learning (FSL) models can be used as LFs in our proposed WS pipeline.
> > * From WS, we build from the idea that we can track the noise in multiple LFs to aggregate labels. We improve upon WS by showing that FSL networks---which obviate the need for users to write their own LFs---enable us to achieve few-shot results with our WS pipeline that exceed competitive baselines.
> > * From FSL, we take ideas for how to train networks with very few labeled images. We improve upon these methods by tracking the noise in each few-shot model.
> >
> > Using simple one-shot models as LFs in our proposed WS segmentation pipeline, we present an end-to-end few-shot segmentation pipeline that achieves, on average, within 4.7 Dice points (range: 1.4 - 8.4) of fully supervised performance and outperforms the next-base baseline by an average 3.6 Dice points across many medical image segmentation tasks, including different imaging modalities, anatomical structures, and pathologies.
> >
> > **Contribution 2: A PGM to aggregate noisy labels for segmentation tasks.**
> >
> > We need to define the graph structure and potentials of a latent-variable PGM to model the unobserved ground truth segmentation and the noisy segmentations output from the LFs. Importantly, the parameters of the PGM need to be learned without access to ground truth labels (if ground truth labels were available, we wouldn’t need WS). Existing WS PGMs rely on estimating the accuracies of LFs [11], however this is not sufficient to model segmentation LFs:
> > * If you naively apply existing WS models to model each LF across all pixels, these models will estimate the global accuracy of each LF. Global accuracy is a problematic metric in segmentation: simply predicting “false” for all pixels in an image can produce global accuracies >99%, and global LF accuracy is not robust to arbitrary image crops (Figure 1 in the submission).
> > * If you attempt to model the accuracy at each pixel in the graph, learning doesn’t scale: there will easily be millions of parameters that need to be learned.
> >
> > We propose a novel PGM, defined in Eqn. (2), to aggregate noisy segmentation LFs. We introduce additional free parameters into this PGM to focus on areas of contention in the segmentation, catching small areas of segmentation LF disagreement without introducing thousands of free parameters to model each edge in the graph.
> >
> > For each dataset, we show we can recover the canonical parameters of the PGM without access to ground truth segmentation masks---i.e. the conditional accuracies of each LF are recovered from the noisy segmentation masks alone---making it useful for the few-shot setting we investigate.
> >
> > **Strengths**
> > 1. We show that our approach outperforms competitive FSL baselines on many segmentation tasks.
> > 2. Our approach is complementary to future methods development in segmentation. New few-shot methods can be used as LFs in Step 1, and more complex segmentation networks, training schedules, or loss functions can be directly substituted into Step 3.
> > 3. As pointed out by R1, “One problem of learning from small datasets is guaranteeing balance and the quality of the training images provided (Changing out even a single image can impact model training).” Our method is suited to handle this challenge. We solve for the parameters of our PGM using the LF outputs, which reflect the particular dataset and labeled training images chosen. If there is a “bad actor” in the small set of labeled images, the impact of that bad actor can be reduced by the PGM.

---

> > > ### Author Response · Authors · 2020-11-22
> > > **Response to all reviewers (4/4): Improved description of methods: new section and figure to describe the workflow.**
> > >
> > > We appreciate the questions about our proposed workflow, and we agree that the description in our original submission should have been more clear. We provide an improved description of our workflow in Section 3.5 and a new Figure 3 in the updated submission to clarify the pipeline for future readers.
> > >
> > > Figure 3 in the updated submission shows a diagram of our method. Our goal is to use a few manually-annotated images and many unlabeled images to train a segmentation network. We propose a stage-wise pipeline that builds upon principles from WS and FSL. Our pipeline proceeds in three steps: (1) noisy training labels are generated by multiple LFs, (2) the noisy labels are aggregated by a PGM, and (3) the aggregated labels are used to train a segmentation model. Each step is described in detail below.
> > > * Step (1). Each one of the five manually annotated images is used to train a one-shot CNN. Because we have five manually annotated images, there are five one-shot CNNs trained. Each one of these one-shot CNNs is an LF which noisily labels all unlabeled training images.
> > > * Step (2). A PGM is used to aggregate the five noisy segmentation masks for each image in the unlabeled training set. The parameters of this PGM are learned for each dataset without using ground truth segmentation masks.
> > > * Step (3): A CNN is trained using the estimated segmentation masks output from the PGM, then fine-tuned with the few hand-labeled images.
> > >
> > >
> > >
> > > We want to again thank the reviewers for their time, comments, and recommendations for improving our paper.
> > >
> > >
> > >
> > >
> > > [1] Cheng Ouyang, Carlo Biffi, Chen Chen, Turkay Kart, Huaqi Qiu, and Daniel Rueckert. Self-supervision with superpixels: Training few-shot medical image segmentation without annotation. ECCV, 2020.
> > >
> > > [2] Krishna Chaitanya, Ertunc Erdil, Neerav Karani, and Ender Konukoglu. Contrastive learning of global and local features for medical image segmentation with limited annotations. NeurIPS, 2020
> > >
> > > [3] Kaixin Wang, Jun Hao Liew, Yingtian Zou, Daquan Zhou, and Jiashi Feng. Panet: Few-shot image semantic segmentation with prototype alignment. ICCV, 2019.
> > >
> > > [4] Abhijit Guha Roy, Shayan Siddiqui, Sebastian Polsterl, Nassir Navab, and Christian Wachinger. ‘Squeeze & excite’ guided few-shot segmentation of volumetric images. Medical image analysis, 2020.
> > >
> > > [5] Krishna Chaitanya, Neerav Karani, Christian F Baumgartner, Anton Becker, Olivio Donati, and Ender Konukoglu. Semi-supervised task-driven data augmentation for medical image segmentation. International Conference on Information Processing in Medical Imaging, 2019.
> > >
> > > [6] Alexander J Ratner, Stephen H Bach, Henry Ehrenberg, Jason Fries, Sen Wu, and Christopher Ré. Snorkel: Rapid training data creation with weak supervision. VLDB, 2018.
> > >
> > > [7] Alexander J Ratner, Christopher De Sa, Sen Wu, Daniel Selsam, Christopher Ré. Data programming: Creating large training sets, quickly. NeurIPS, 2016.
> > >
> > > [8] Stephen H. Bach, Daniel Rodriguez, Yintao Liu, Chong Luo, Haidong Shao, Cassandra Xia, Souvik Sen, Alex Ratner, Braden Hancock, Houman Alborzi, et al. Snorkel drybell: A case study in deploying weak supervision at industrial scale. International Conference on Management of Data, 2019.
> > >
> > > [9] Ying Sheng, Nguyen Vo, James B. Wendt, Sandeep Tata, and Marc Najork. Migrating a privacy-safe information extraction system to a software 2.0 design. Innovative Data Systems Research, 2020.
> > >
> > > [10] Jared A Dunnmon, Alexander J Ratner, Khaled Saab, Nishith Khandwala, Matthew Markert, Hersh Sagreiya, Roger Goldman, Christopher Lee-Messer, Matthew P Lungren, Daniel L Rubin, et al. Cross-modal data programming enables rapid medical machine learning. Patterns, pp. 100019, 2020.
> > >
> > > [11] Daniel Y Fu, Mayee F Chen, Frederic Sala, Sarah M Hooper, Kayvon Fatahalian, and Christopher Ré. Fast and three-rious: Speeding up weak supervision with triplet methods. ICML, 2020

---

### Decision · Program_Chairs · 2021-01-07
**Final Decision**

**Decision:**

Accept (Poster)

**Comment:**

This article proposes a weakly supervised few-shot learning method for medical imaging segmentation. While initially, the article presented several problems indicated by the reviewers, e.g., the explanation of the novelty and contributions, the explanation of the method, and the experimental evaluation, the authors made a great effort addressing most of the reviewers' comments and uploaded an updated version of the article. However, still, the evaluation part of the article is a bit weak. But the article contains interesting contributions. Accordingly, I recommend accepting the paper at ICLR2021.